# Circulating re-entrant waves promote maturation of hiPSC-derived cardiomyocytes in self-organized tissue ring

Junjun Li [1,11], Lu Zhang[2,11], Leqian Yu[3], Itsunari Minami[4], Shigeru Miyagawa[1], Marcel Hörning [3,10], Ji Dong[5], Jing Qiao[3], Xiang Qu[1], Ying Hua[1], Nanae Fujimoto[1], Yuji Shiba[6], Yang Zhao[7], Fuchou Tang[5], Yong Chen[3,8], Yoshiki Sawa[1✉], Chao Tang [2✉] & Li Liu[1,9✉]

Directed differentiation methods allow acquisition of high-purity cardiomyocytes differentiated from human induced pluripotent stem cells (hiPSCs); however, their immaturity characteristic limits their application for drug screening and regenerative therapy. The rapid electrical pacing of cardiomyocytes has been used for efficiently promoting the maturation of cardiomyocytes, here we describe a simple device in modified culture plate on which hiPSC-derived cardiomyocytes can form three-dimensional self-organized tissue rings (SOTRs). Using calcium imaging, we show that within the ring, reentrant waves (ReWs) of action potential spontaneously originated and ran robustly at a frequency up to 4 Hz. After 2 weeks, SOTRs with ReWs show higher maturation including structural organization, increased cardiac-specific gene expression, enhanced $Ca^{2+}$-handling properties, an increased oxygen-consumption rate, and enhanced contractile force. We subsequently use a mathematical model to interpret the origination, propagation, and long-term behavior of the ReWs within the SOTRs.

[1] Department of Cardiovascular Surgery, Osaka University Graduate School of Medicine, 2-2 Yamadaoka, Suita, Osaka 565-0871, Japan. [2] Center for Quantitative Biology and Peking-Tsinghua Center for Life Sciences, Academy for Advanced Interdisciplinary Studies, Peking University, 100871 Beijing, China. [3] Institutes for Integrated Cell-Material Sciences (WPI-iCeMS), Kyoto University, Yoshida-Ushinomiya-cho, Sakyo-ku, Kyoto 606-8501, Japan. [4] Department of Cell Design for Tissue Construction Faculty of Medicine, Osaka University, Osaka 565-0871, Japan. [5] Biomedical Pioneering Innovation Center, College of Life Sciences, Peking University, 100871 Beijing, China. [6] Department of Regenerative Science and Medicine, Institute for Biomedical Sciences, Shinshu University, 3-1-1 Asahi, Matsumoto, Nagano 390-0821, Japan. [7] State Key Laboratory of Natural and Biomimetic Drugs, The MOE Key Laboratory of Cell Proliferation and Differentiation, Institute of Molecular Medicine, Peking-Tsinghua Center for Life Sciences, Peking University, 100871 Beijing, China. [8] PASTEUR, Département de chimie, école normale supérieure, PSL Research University, Sorbonne Universités, UPMC Université Paris 06, CNRS, Paris 75005, France. [9] Department of Drug Discovery Cardiovascular Regeneration, Osaka University Graduate School of Medicine, 2-1 Yamadaoka, Suita, Osaka 565-0871, Japan. [10]Present address: Institute of Biomaterials and Biomolecular Systems, University of Stuttgart, 70569 Stuttgart, Germany. [11]These authors contributed equally: Junjun Li, Lu Zhang. ✉email: sawa-p@surg1.med.osaka-u.ac.jp; tangc@pku.edu.cn; li-liu@surg1.med.osaka-u.ac.jp

Although human-induced pluripotent stem cell (hiPSC) derived cardiomyocytes have been proposed as an abundant resource for tissue engineering, drug screening, and regenerative-medicine applications[1], they exhibit characteristics different from adult human cardiomyocytes, including immature sarcomere structure and morphology, a fetus-like gene expression profile, and inadequate $Ca^{2+}$-handling properties[2–4]. The immature nature of hiPSC-derived cardiomyocytes potentially hinders their reflection of adult heart physiology for disease modeling and drug assessment.

To achieve higher degrees of hiPSC-derived cardiomyocytes maturation and function, a variety of methods have been developed, including dynamic culture[5], three-dimensional (3D) engineered heart tissue[6–8], 3D printing[9], addition of factors[10], extracellular matrix[10,11], and external stimulation. Electrical stimulation, especially high-frequency pacing, has long been suggested as an effective method for maturing muscle cells[3,12–15], which upregulates cardiac markers[13,14,16], enhances $Ca^{2+}$-handling properties[3], and promotes cardiomyocyte alignment[17]. During this process, appropriate stimulation protocols are required to minimize possible tissue damage[18]. Additionally, it remains challenging to rapidly pace cardiomyocyte tissues (≥2 Hz) over long time[19] due to possible side effects, such as heavy metal poisoning, electrolysis, pH shift, and the generation of reactive oxygen species (ROS)[20,21]. Moreover, upscaling for mass stimulation is either difficult or requires high level of power consumption[19]. As an important complement, mechanical stimulation can promote cardiomyocyte maturation, with cyclic mechanical stress applied by external stretching devices capable of forcing muscle-cell assembly into structurally and functionally aligned 3D syncytium[22], thereby affecting their gene expression[23–25] and $Ca^{2+}$ cycling[26].

On the other hand, spiral waves [re-entrant waves (ReWs)] within cardiac tissue have long been investigated as a model for arrhythmia studies[27–31], and the rapid beating caused by in vivo spiral wave could lead to dysfunction of heart and even death of a patient. However, when used in an in vitro condition, the rapid beating caused by spiral wave, similar to rapid electrical stimulation[14], could be beneficial for cardiomyocytes' maturation.

In this study, we create a platform capable of promoting rapid formation of hiPSC-derived cardiomyocytes into 3D self-organized tissue rings (SOTRs), where propagation of an action potential in the form of ReWs can spontaneously originate and travel around the closed-loop circuit, thereby making the cardiomyocytes beat at a high frequency (~2–4 Hz) continuously and robustly up to more than 89 days without any external stimulation. Additionally, we found that the beating frequency and the ReW speed can be adjusted by changing the diameter of the ring. Furthermore, we construct a mathematical model in order to elucidate the origination, propagation, and long-term behavior of the ReWs, with the model ultimately agreeing well with the experimental data. After 2 weeks of culture, the SOTRs with ReWs demonstrate improved structural organization, upregulated cardiac-specific gene expression, enhanced $Ca^{2+}$-handling properties, an increased oxygen-consumption rate (OCR), and enhanced contractile force. Our results demonstrate, to our knowledge, a novel approach for spontaneous cardiomyocyte maturation and could serve as an economical and practical system for future production of mature hiPSC-derived cardiomyocytes.

## Results

### Self-organization of a hiPSC-derived cardiomyocyte ring. We created 3D SOTRs by plating hiPSC-derived cardiomyocytes in a culture dish with a pillar in the center, around which the

cardiomyocytes aggregated and formed a thick tissue ring within 2 days (432.72 ± 56.18 μm on day 2, Fig. 1a–c and Supplementary Fig. 1). As indicated by the genetically encoded calcium indicator GCaMP3, we found looped activation propagations within the SOTRs (i.e., ReWs; Supplementary Movie 1), with zero to two ReWs present in one SOTR (Fig. 1d, e and Supplementary Movie 2). Besides circular geometry, other geometries have also been tested, such as 4-point star, 5-point star (Supplementary Fig. 2a), and the geometry between rectangular and circular could also be obtained. However, the formation of the tissues throughout different areas of these templates was less homogeneous than that of the tissues in the circular geometry. In addition, the circular ring tissues were more equally and stably organized while maintaining the ReWs with a higher ratio (Supplementary Fig. 2b). Thus, we chose the circular ring to engineer cardiac tissues. Fluorescence-activated cell sorting (FACS) (Supplementary Fig. 1b), voltage dye staining, and immunostaining data indicated that the SOTRs were mostly ventricular cardiomyocytes and few fibroblast cells (Supplementary Figs. 3 and 4).

A trace recording (Fig. 1d, e) indicated no long rest periods between successive ReWs, resulting in higher beating rates in SOTRs capable of sustaining a higher number of ReWs (Fig. 1e, f). The spontaneous beating rate of SOTRs with zero ReW was 0.21 ± 0.10 Hz on day 6 and remained stable for 2 weeks, whereas the rates of SOTRs with one or two ReWs on day 6 were much higher (2.40 ± 0.49 and 3.30 ± 0.39 Hz, respectively). Notably, the wave speed in the ReW groups was much lower (~2 cm s$^{-1}$) than that of the spontaneous beating group (0 ReW; 5.93 ± 1.50 cm s$^{-1}$). It is possible that the slower speed together with the shortened pacing interval and refractory period might have been caused by the higher beating frequency in the ReW groups relative to that in the zero ReW group (Supplementary Fig. 5 and Supplementary Movie 3), which agrees with previous reports associated with excitable media[32–36]. Additionally, after the ReW was stopped, the speed of spontaneous beating in the two ReW group increased significantly ($p = 0.01$) to more than 10 cm s$^{-1}$ (Fig. 1g).

The beating rates of the one and two ReW groups increased slightly over 2 weeks of culturing, reaching 3.03 ± 0.68 and 3.92 ± 0.69 Hz on day 14, respectively. After a 6-day culture period, 83.8% of 204 SOTRs had one or two ReWs, 14.2% had zero ReW, and 2% had three ReWs; however, these percentages changed after 2 weeks of culture to 28.5% with zero ReW, 48.9% with one ReW, 22.6% with two ReWs, and no SOTRs with three ReWs (Fig. 1h). Noise and disturbances, such as those during medium changing, might have provoked changes in and/or disappearance of ReWs. In the long-term culture, we found that ReWs could be maintained in SOTRs for >89 days (Supplementary Fig. 6). In addition to the static culture, dynamic culture of SOTRs on a rotary shaker has been applied to check the effect on ReWs. During the culture, we did find the improvement of beating frequency of cardiomyocytes in the ring with ReWs (Supplementary Fig. 7), which significantly increased from 3.30 ± 0.39 to 3.89 ± 0.18 Hz at day 6 ($p = 0.038$) and from 3.92 ± 0.69 to 5.57 ± 0.06 Hz at day 14 ($p = 0.001$). The higher beating frequency has been previously reported to be associated with higher maturation of cardiomyocytes[14]. On another hand, there are no significant differences between the zero ReW groups with or without rotary culture. These data indicated dynamic culture could improve the availability of oxygen and glucose to cardiomyocytes in the ReW group[5]. Despite the improvement, we also found that the ReW occurrence ratio in the dynamic group dropped to 50% on day 6 and 40% on day 14; the disappearance of the ReW was perhaps due to the disturbance caused by the dynamic medium

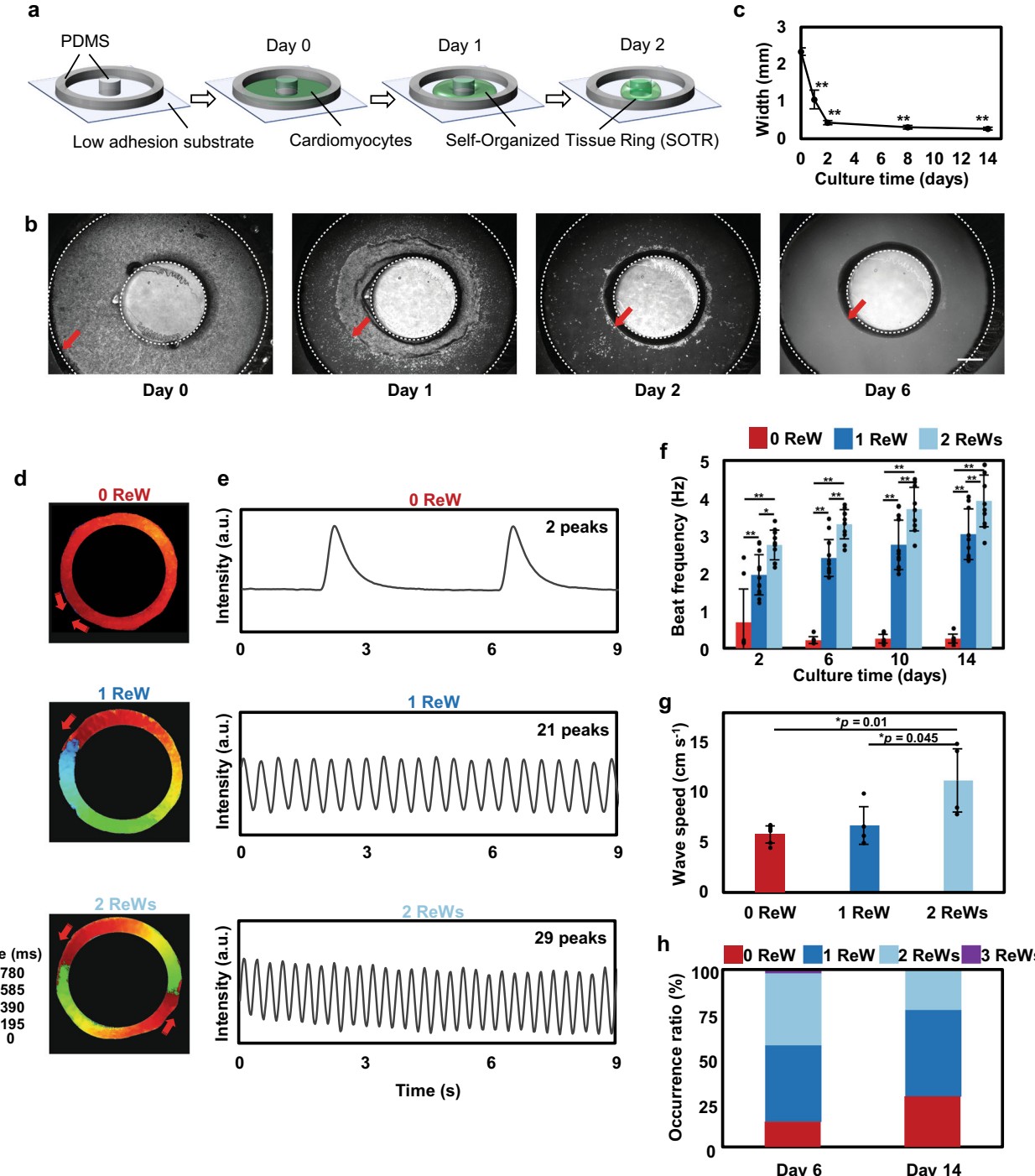

**Fig. 1 ReWs promoted rapid beating of cells in SOTRs. a** Schematic describing SOTR formation. **b** Bright-field images of hiPSC-derived cardiomyocytes in the template. The red arrows indicate the edge of cardiac tissues in the template, and the dashed lines indicate the PDMS block and pillar boundaries, respectively. Pillar diameter = 3 mm. Scale bar represents 800 μm. **c** Quantification of the width of the SOTRs on the indicated culture day (mean ± s.d.; $n$ = 4 biologically independent samples). **$P < 0.001$ (ANOVA). **d** Activation map of GCaMP3-positive SOTRs with zero, one, or two ReWs. The red arrows indicate the propagation direction of the action potential. **e** GCaMP3 fluorescence signal at a fixed position on the ring of SOTRs with zero, one, or two ReWs on day 6. **f** Beat rates of SOTRs at different culture times (mean ± s.d.; 2 ReWs: $n$ = 10; 1 ReW: $n$ = 12; 0 ReW: $n$ = 8 biologically independent samples). *$P < 0.05$; **$P < 0.001$ (ANOVA). **g** The wave speed of spontaneous beating in SOTRs with zero, one, or two ReWs after 14 days of culture; for ReW groups, the speed was recorded after the ReWs are stopped and the spontaneous beating was recovered (mean ± s.d.; 2 ReWs: $n$ = 4; 1 ReW: $n$ = 4; 0 ReW: $n$ = 6 biologically independent samples). *$P < 0.05$ (ANOVA). **h** The percentage of occurrence for different numbers of ReWs on days 6 ($n$ = 204 biologically independent samples) and 14 ($n$ = 186 biologically independent samples), respectively.

flow. More future research work is necessary to improve the culture well design and reduce the disturbance of medium.

To better understand the nature of these ReWs according to their organization, propagation, and long time behavior, we constructed a mathematical model (Supplementary Information and Supplementary Fig. 8) comprising a ring of cells, each of which can beat spontaneously with an intrinsic frequency. The total number of the cells in each ring in the model is proportional to the diameter of the ring. The coupling between neighboring cells through gap junctions[37,38] gradually increases with time in order to simulate the self-organization and formation processes of the SOTR. Initially, all cells were independently set to a random phase of beating, with all cells beating independently. As the gap junctions form and strengthen, the beating of one cell harbors an increasing probability of triggering beating in its neighboring cell, thereby forming a propagating wave. Initially, these waves are unsynchronized and short lived, as they initiate, disappear, and sometimes meet and collide (Supplementary Movie 4). Over time, a SOTR is left with one dominant propagating-wave mode comprising zero or more ReWs (Fig. 2a and Supplementary Movie 4), a process similar to that observed in our experimental findings (Supplementary Movie 5). Notably, SOTRs with one or two ReWs accounted for most of the simulation samples, a trend also observed in our experimental results (Fig. 2b).

We then investigated how the ReW features and properties were affected by characteristics, such as ring diameter, in the mathematical model and experiments. The beating frequency of cells in SOTRs decreased with increasing ring diameter, as shown with one ReW in Fig. 2c. We noticed that a constant ReW speed implied a linear decrease in beating frequency with ring diameter (which is proportional to the ring perimeter), and that the greater the perimeter, the more time the wave would spend to travel around it. However, the data in Fig. 2c show a slower-than-linear decrease; therefore, we measured the wave speed, finding that it increased along the ring diameter in both the simulation and experiment (Fig. 2d). As discussed in the Supplementary Information, this was due to the ability of a single hiPSC-derived cardiomyocyte to spontaneously beat. Cells in larger rings would wait longer for the next ReW to arrive, thereby making them easier to be activated by the wave front and resulting in a faster wave speed. This insight allowed us to predict the speed of two and three ReWs. As shown in Fig. 2e, the prediction agreed well with our experimental results. Additionally, the mathematical model showed that the maximum number of ReWs that a SOTR could sustain increased along with ring diameter, which was confirmed experimentally (Fig. 2f). Here, the mathematical model predicted that SOTRs with a diameter of 17 mm could contain as many as 19 ReWs. Experimentally, we found that SOTRs with a 3-mm pillar were optimal based on their highest occurrence of ReWs (~90% at ≥1 ReW) (Supplementary Fig. 9), which was probably caused by the passive stretch in the SOTRs with a different pillar diameter and its effect on the anisotropic structural organization. Moreover, the beating frequency of one ReW was only slightly slower than that in SOTRs with a 1-mm pillar; therefore, SOTRs with a 3-mm pillar were chosen for subsequent investigations.

**ReWs are associated with maturation of cardiomyocytes**. The human fetal heart rate varies, although it generally stabilizes at ~3 Hz, whereas the adult human heart rate is ~1 Hz[39]. Electrical stimulation has been used to mature cardiomyocytes by pacing their beating at a certain higher-than-normal frequency relative to a normal human rate[3]; however, continuous and high electrical stimulation can cause cell damage. Here, ReWs within the SOTRs were able to make the cardiomyocytes

beat at various frequencies in the absence of external stimulation.

We used RNA-sequencing to compare the gene expression profiles among different groups. The principal component analysis (PCA; Fig. 3a) and hierarchical clustering of Spearman's correlation results (Fig. 3b) showed closer correlations between hiPSC-derived cardiomyocytes receiving training by ReWs as compared with those without ReW training. Additionally, we found different gene expression patterns among zero, one, and two ReW groups (Fig. 3c). Functional annotation according to Gene Ontology (GO) analysis revealed that the upregulated genes (adjusted $P < 0.05$; fold change >1.5) in SOTRs with two ReWs, compared to those with no ReWs, were related to the response to unfolded protein and a number of maturation-related terms. Most of the GO-enriched terms were associated with cardiac development, including muscle-structure development, extracellular-structure organization, actin-filament-based processes, tissue morphogenesis, and muscle-system processes. The response to unfolded protein is a coping response to mitigate or eliminate endoplasmic reticulum (ER) stress caused by hypoxia[40], which could result from the high-frequency beating of the SOTRs with ReWs. To verify these findings, we performed immunostaining and quantitative polymerase chain reaction (qPCR) analysis. Immunostaining for cardiac-specific markers clearly revealed higher expression of β-myosin heavy chain (β-MHC), a cardiac maturity marker correlated with contractile velocity, within SOTRs with one and two ReWs as compared with levels observed in the zero ReW group (Fig. 3e, f). Moreover, analysis of messenger RNA (mRNA) expression revealed a number of upregulated genes, including *MYH7* (encoding β-MHC) and genes involved in sarcomere structures (*ACTN2* and *DES*), ventricular structures (*MYL2* and *MYL3*), ER-Ca$^{2+}$ function (*PLN*), myoglobin (*MB*), and β1-adrenoceptor (*ADRB1*). SOTRs with ReWs had higher-fold upregulated genes as compared with those with zero ReW (Fig. 3e–g) Owing to the higher beating rates during culture, which agreed well with previous reports utilizing electrical stimulation at different frequencies[3,16].

When the RNA-sequencing data of SOTR cardiomyocytes was compared with those of adult heart cells and hiPSC-derived cardiomyocytes from a previous report[41] (Supplementary Fig. 10), the SOTR cardiomyocytes showed lower nodal- (*TBX18*, *SHOX2*, *MSX2*, *TBX2*, *HCN1*, *HCN4*) and atrial-related (*MYH6*, *NPPA*) and higher ventricular-related (*MYL2*) gene expression than the previously reported cardiomyocytes[41]. Both the SOTR cardiomyocytes and cardiomyocytes of the previous study showed lower expression of *MYH7* and *MYL2*, but higher *IRX4* expression than the adult cardiomyocytes. In addition, after ReW training, the gap junction (*GJA1*) and the n-cadherin (*CDH2*) in two ReW group was higher than that in zero ReW group (Supplementary Figs. 11 and 12).

Hematoxylin and eosin (HE) staining showed cardiomyocytes with ReWs densely packed with each other to a greater degree than those without ReWs (Supplementary Fig. 13). Moreover, SOTRs with ReWs demonstrated densely packed cardiac myofilaments along the ring orientation, which was also along the path of the ReWs, whereas cardiomyocytes in SOTRs without ReWs (0 ReW) were randomly oriented and poorly organized (Fig. 4a, b). Additionally, after a 14-day culture period, cardiomyocytes in SOTRs with ReWs strongly expressed the α-actinin as compared with those in SOTRs without ReWs (Fig. 4c), which agrees with the gene expression results (*ACTN2*, Fig. 3g). Importantly, cardiomyocytes in SOTRs with ReWs exhibited significantly longer sarcomere length than those without ReWs (1 ReW: 1.71 ± 0.08 μm; 2 ReWs: 1.83 ± 0.10 μm; and 0 ReW: 1.49 ± 0.08 μm; 1 ReW vs. 0 ReW: $p = 0.012$; 2 ReWs vs. 0 ReW: $p = 0.001$) (Fig. 4c, d), which is closer to the sarcomere length in

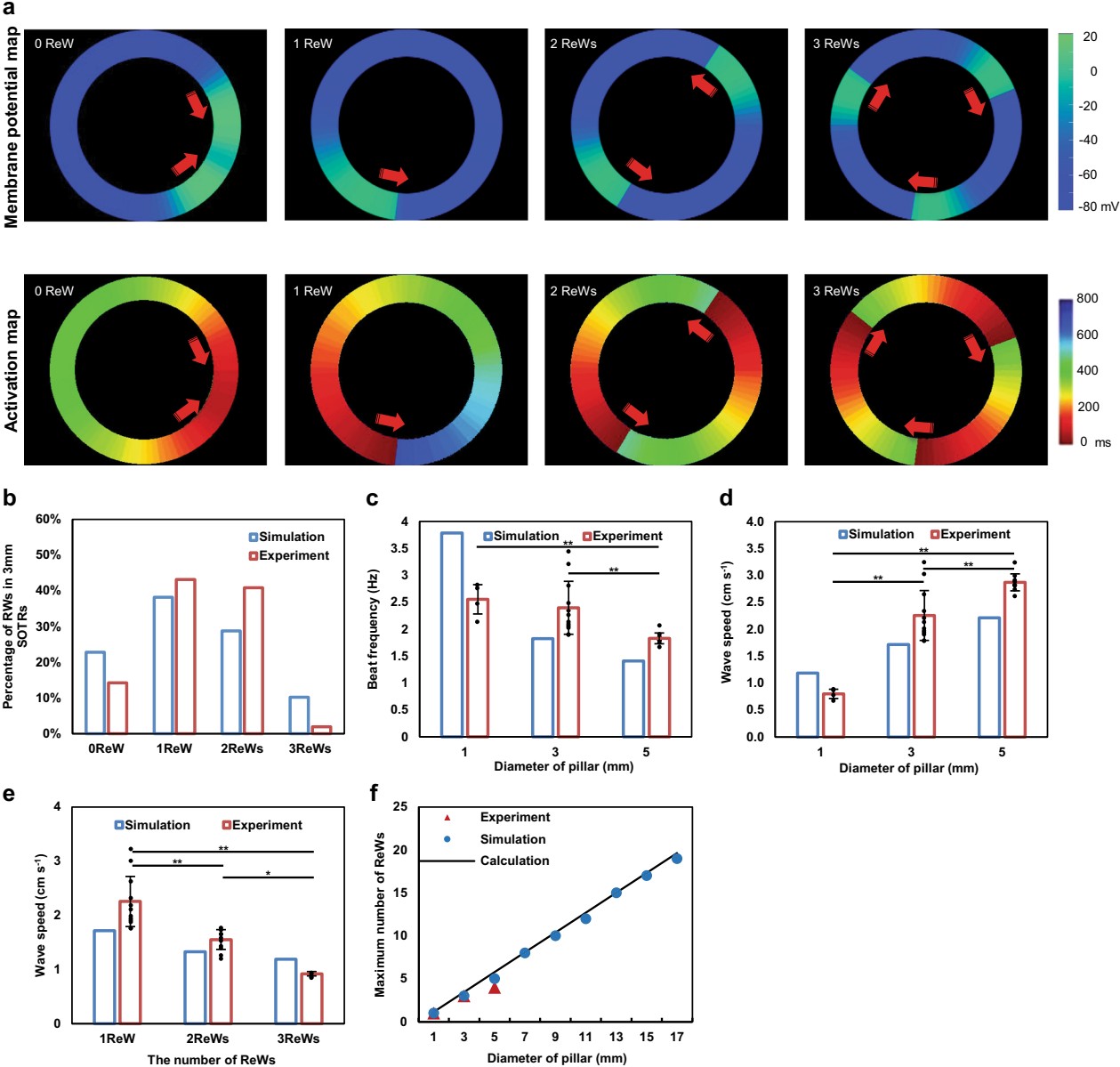

**Fig. 2 Accurate reproduction of the ReW features in SOTRs using a mathematical model. a** Examples of simulation results of the model (upper panel) and the activation map (lower panel). For the group with zero ReW, an action potential initiates from the left position of the ring and propagates in the opposite directions along the ring, after which the two waves meet and annihilate on another. The process starts again with new waves initiating at the same position periodically. For groups with one, two, or three ReWs, stable ReWs travel around the ring. Membrane potential and activation time are color coded. **b** The percentage of stable ReWs in SOTRs at day 6 (diameter = 3 mm; $n = 60$ in the simulation and $n = 204$ biologically independent samples in the experiment). **c** Beat rate (Hz) and **d** ReW speed in SOTRs with one ReW at day 6 for different pillar diameters [experiment: 1-mm SOTR ($n = 4$); 3-mm SOTR ($n = 12$); 5-mm SOTR ($n = 10$ biologically independent samples)]. **$P < 0.001$ (ANOVA). **e** Speeds of one, two, or three ReWs in SOTRs with a pillar diameter of 3 mm [experiment: mean ± s.d.; 3 ReWs ($n = 4$); 2 ReWs ($n = 10$); 1 ReW ($n = 12$ biologically independent samples)]. *$P < 0.05$; **$P < 0.001$ (ANOVA). **f** The maximum number of ReWs in SOTRs with different pillar diameters.

human adult cardiomyocytes[42]. Moreover, on day 2 the SOTRs with or without ReWs all demonstrated randomly distributed and poorly organized cardiomyocytes, and similar maturation level (Supplementary Fig. 4). ReW could also be induced in zero ReW SOTR by rapid electrical stimulation at day 14 (Supplementary Movie 6). Collectively, these data, together with the previous day 14 results, indicate that that the ReW pacing are associated with improved maturation.

Furthermore, electron microscopy indicated that cells in the ReW groups exhibited larger sarcomeric bundles, well-defined Z disks, A-bands, and myofibrils (Fig. 4e). These data suggest that cardiomyocyte maturation was improved within the SOTRs with

ReWs compared to those with zero ReW. Our findings confirmed the genetic and structural maturation of SOTRs with ReWs.

**ReWs are associated with improved Ca²⁺-handling properties**. An extracellular flux analyzer was used to characterize mitochondrial function in cardiomyocytes within SOTRs. To record the maximum activity of the electron-transport chain from adenosine triphosphate (ATP) synthesis, mitochondrial ATP synthase was inhibited by oligomycin, and a proton-gradient discharger was added to measure the maximum mitochondrial respiration. Our results indicated that the maximum respiration

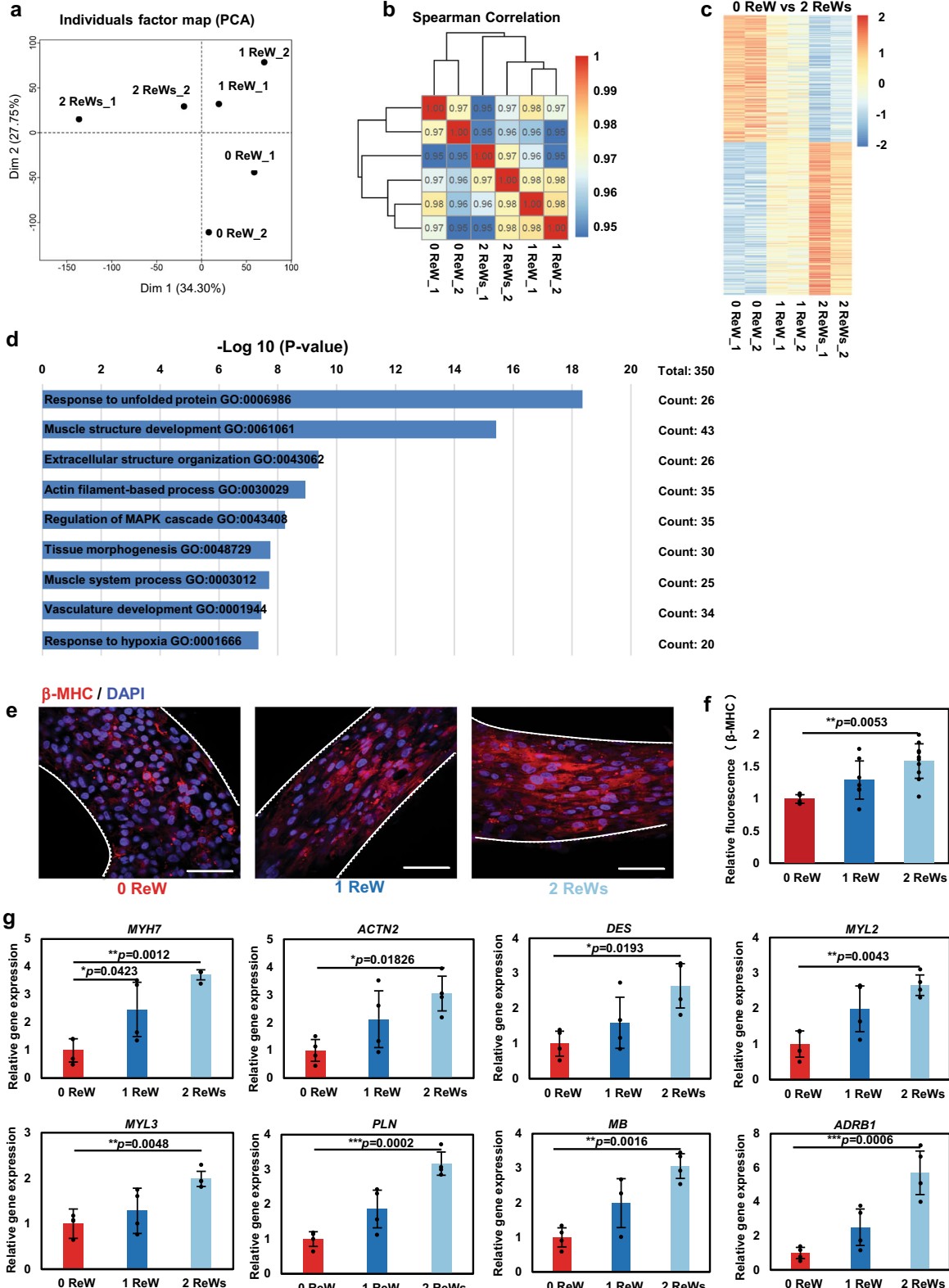

rate of groups with two ReWs was significantly higher than that observed in groups with zero or one ReW (Fig. 5a, b; 2 ReWs vs. 1 ReW: $p = 0.0075$; 2 ReWs vs. 0 ReW: $p = 0.001$), indicating that a higher number of ReWs led to increased mitochondrial activity.

Because human cardiomyocytes rely on $Ca^{2+}$ influx from the sarcoplasmic reticulum (SR) for contraction, we investigated the SR function by applying caffeine to SOTRs in order to open the

SR ryanodine channels[3]. Although all groups responded to the addition of 5 mM caffeine (Fig. 5c, d), SOTRs with two ReWs exhibited significantly higher intensity changes than the groups with zero ReW, indicating a higher maturation level in $Ca^{2+}$-handling properties ($p = 0.0237$). Additionally, we recorded the contractile force of SOTRs using a mechanical tester (Fig. 5e, f and Supplementary Fig. 14), which revealed that SOTRs with

**Fig. 3 ReWs are associated with the upregulation of cardiac-specific gene expression. a** PCA of SOTRs with or without ReWs based on RNA-sequencing data. **b** Heatmap showing hierarchical clustering of the correlation matrix resulting from comparison of expression values for each sample. Correlations calculated using Spearman's correlation. **c** Heatmap showing the relative expression levels (z-score) of 349 genes from SOTRs with ReWs (1 and 2 ReWs) as compared with SOTRs with zero ReW. **d** GO categories significantly enriched for genes upregulated in SOTRs with two ReWs as compared with SOTRs with zero ReW. Genes with fold changes >1.5 and a $P < 0.05$ were analyzed. The enriched terms are listed. **e** Representative confocal images of SOTRs with zero, one, or two ReWs on day 14. Cardiomyocytes were stained with anti-β-MHC (red) and DAPI (blue). Scale bar represents 50 μm. **f** Quantification of β-MHC levels in SOTRs on day 14. The total integrated fluorescence of β-MHC acquired from confocal imaging was normalized to that of SOTRs with zero ReW (mean ± s.d.; $n = 4$ biologically independent samples). $**P < 0.01$ (ANOVA). **g** The expression of cardiac-specific genes (qPCR) in SOTRs 14 days after cell seeding (mean ± s.d.; $n = 4$ biologically independent samples). $*P < 0.05$; $**P < 0.01$; $***P < 0.001$ (ANOVA).

either zero or two ReWs generated active forces that increased with the applied stretching force in a Frank–Starling-like fashion[43]. Moreover, the group with two ReWs generated forces with larger amplitude than that in the group with zero ReW (2 ReWs: $0.54 \pm 0.15$ mN mm$^{-2}$; 0 ReW: $0.23 \pm 0.12$ mN mm$^{-2}$).

## Discussion

Excitation–contraction coupling is important for cardiomyocyte development and function[44], and beating rate might play an important role in cardiac maturation. The human fetal heartbeat increases linearly during embryonic development from ~1 to ~3 Hz within 9 weeks in utero[45,46]. hiPSC-derived cardiomyocytes functionally and morphologically resemble fetal cardiomyocytes; however, the average beating rate of hiPSC-derived cardiomyocytes is much lower than that of human fetal cardiomyocytes. Previous studies promoted cardiomyocyte maturation by increasing their beating rates; however, the high beat rates of hiPSC-derived cardiomyocytes could only be achieved by applying high-frequency electrical stimulation[3,14,16], making the sustainable application of such stimuli technically challenging. Moreover, electrical stimulation may cause potential side effects, including pH shift, ROS generation, electrolysis[20], and cell damage[18].

In the present study, we created SOTRs with spontaneously generated ReWs capable of sustainably causing cardiomyocytes to beat at high frequencies (up to 4 Hz) in the absence of external stimulation. We found that cardiomyocytes with ReWs demonstrated dramatically improved structural maturation, enhanced cardiac gene expression, and Ca$^{2+}$-handling properties in a frequency-dependent manner. Because this activity might also cause hypoxia in SOTRs, in the future, an enhanced oxygen supply, such as that afforded by dynamic culture[5], can be utilized during SOTR culture. Additionally, although the cardiomyocytes trained by ReWs remained less mature than those generated by state-of-the-art methods[11,14,47], the maturation level in ReWs group could be further affected by changing the stimulation window/duration and improving the beating frequency.

SOTRs were formed through a one-step, gel-free method utilizing a polydimethylsiloxane (PDMS) mold and commercially available Petri dishes. SOTR formation required <2 days, which is a much shorter period than the previously reported tissue ring formation time (~7–14 days)[22,48,49], owing to multiple factors, including the use of gel-free medium, the low attachment surface of the culture dish, and the ability of the cardiomyocytes to aggregate into a 3D construct. This methodology could be an important complement to current widely used methods[3,20,22,26], and the rapid formation of the tissue ring without any exogenous ECM, such as collagen or fibrin, might provide new insights into future mass production of mature tissues for drug screening and regenerative applications[50]

Given the large cell number required for SOTR formation in one well ($4 \times 10^5$ cells), it remains difficult to achieve a high-throughput assessment that is comparable to that of a previous study where 200 tissue constructs were created with per million

cardiac cells[51]. Further optimizations are needed to scale down the PDMS mold and reduce the plated cell number. Additionally, because force generation is the primary function of cardiomyocytes, force quantifications have been recently utilized to assess drug-related effects[51,52]; therefore, future mold design for SOTR formation might include a force transducer[53], such as an elastic silicone pillar, capable of offering multiple parameters for more relevant assessment of drug response to pharmacologic agents. Furthermore, since the electrical pacing could be controlled with different loading levels (by gradually increasing or decreasing frequency, or by tuning on/off the region of stimulation over time)[14,51], it would be useful to develop SOTRs with controllable stimulation. SOTRs with controllable silicone pillar could be used to regulate the frequency of ReWs in real time.

It remains an open question as to what degree of cardiomyocyte maturation is necessary for regenerative applications[54], such as injection into or engraftment[50] onto an infarcted heart. The various maturation levels of SOTRs under different ReW training levels might offer preconditioned cardiomyocytes for delivery into an infarcted myocardium, thereby allowing studies on how cardiomyocyte maturation affects therapeutic efficacy.

The beating frequency and sarcomere length of ReW-trained cardiomyocytes did not increase with culture time (>89 days) longer than 14 days (Supplementary Fig. 6). This indicated that the rapid pacing maturation process could be limited within a specific time window, similar to the findings of a previous report[14]. Moreover, because the frequency of the ReWs within the SOTRs can be adjusted by changing the SOTR diameter, and given that the ReWs could be sustained for >89 days, it might be possible to create a pacemaker tissue with an adjustable beat rate for use as an in vitro model for drug assessment or potentially for in vivo heart pacing.

In conclusion, we found that ReWs could be spontaneously generated and maintained within a SOTR comprising hiPSC-derived cardiomyocytes, and that the ReWs were able to make the cardiomyocytes beat at a high frequency comparable to that found in utero during embryonic development. Moreover, the ReWs are associated with structural and functional maturation of the cardiomyocytes, thereby offering a supplementary approach to electrical stimulation-based maturation of electrically active cell types.

## Methods

**Differentiation and culture of hiPSC-derived cardiomyocytes.** GCaMP3-positive hiPSCs (253G1) were cultured and differentiated according to previously published methods[55,56]. All experiments involving the use of hiPSCs were performed following Kyoto University and Osaka University guidelines. After 30 to 50 days of differentiation, cardiomyocyte colonies floating in the medium were collected and dissociated into a single-cell suspension by stirring for between 1 and 2 h in protease solution [0.1% collagenase type I, 0.25% trypsin, 1 U mL$^{-1}$ DNase I, 116 mM NaCl, 20 mM HEPES, 12.5 mM NaH$_2$PO$_4$, 5.6 mM glucose, 5.4 mM KCl, and 0.8 mM MgSO$_4$ (pH 7.35)][56]. The cardiomyocyte purity was characterized with flow cytometry and the cardiomyocytes with high purity (>85%) were used for following experiments.

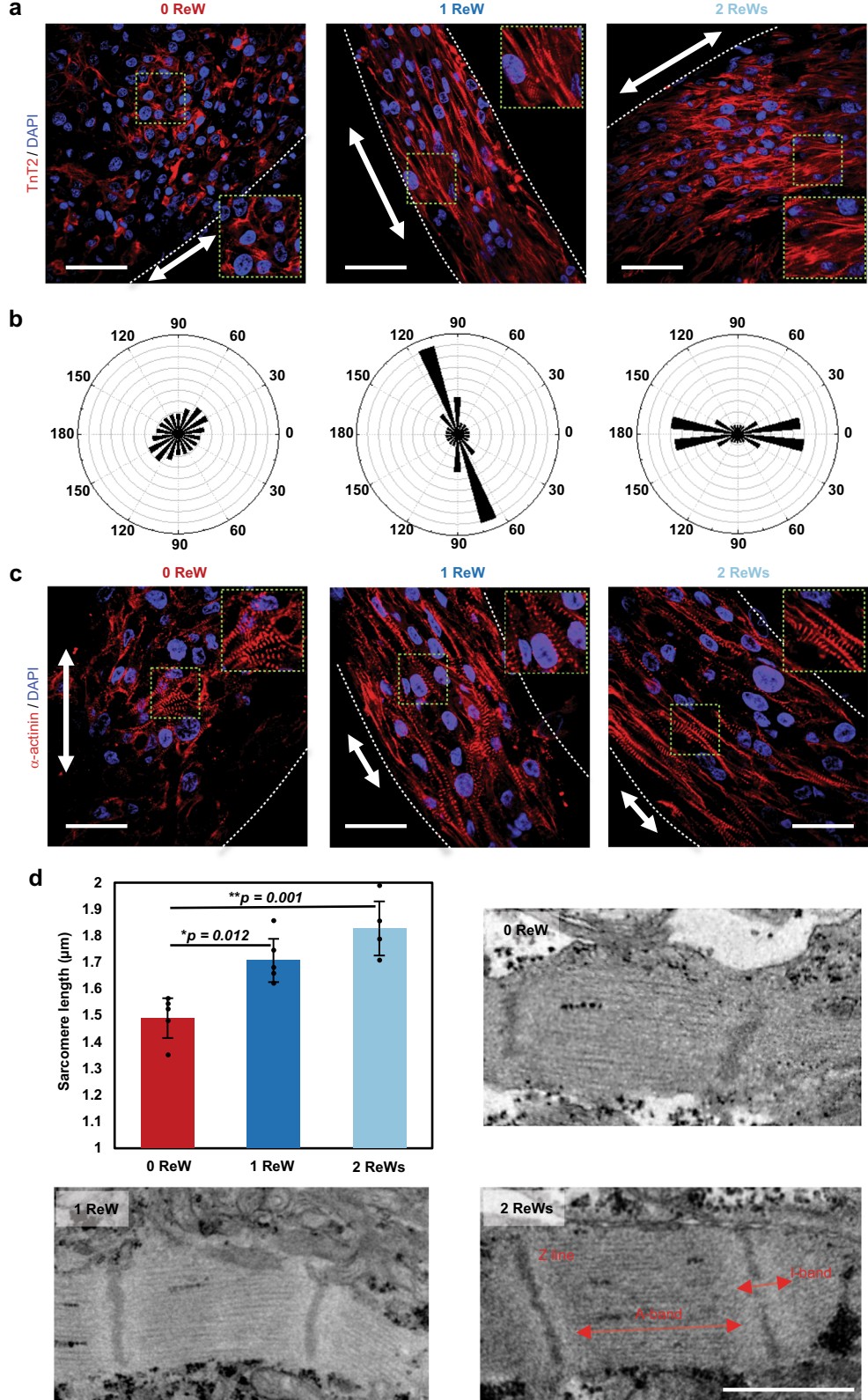

**Fig. 4 ReWs are associated with cardiomyocyte alignment. a** Representative confocal images of SOTRs with zero, one, or two ReWs. cardiomyocytes were stained with anti-TnT2 (red) and DAPI (blue), revealing cardiomyocyte alignment in ReW samples at day 14. Arrows mark the ring orientation, and the area delimited by a green dashed square is zoomed-in in each confocal image. Scale bar represents 50 µm. **b** Angular graph based on Fourier component analysis of TnT2 orientation distribution. **c** Representative confocal images of stained SOTRs with different ReWs (red: α-actinin; blue: DAPI). Arrows mark the ring orientation at day 14. Scale bar represents 30 µm. **d** Sarcomere lengths of cardiomyocytes within SOTRs on day 14 [mean ± s.d.; 0 ReW ($n = 5$); 1 ReW ($n = 5$); 2 ReWs ($n = 4$ biologically independent samples)]. *$P < 0.05$; **$P < 0.01$ (ANOVA). **e** TEM analysis of SOTRs on day 14. Scale bar represents 1 µm.

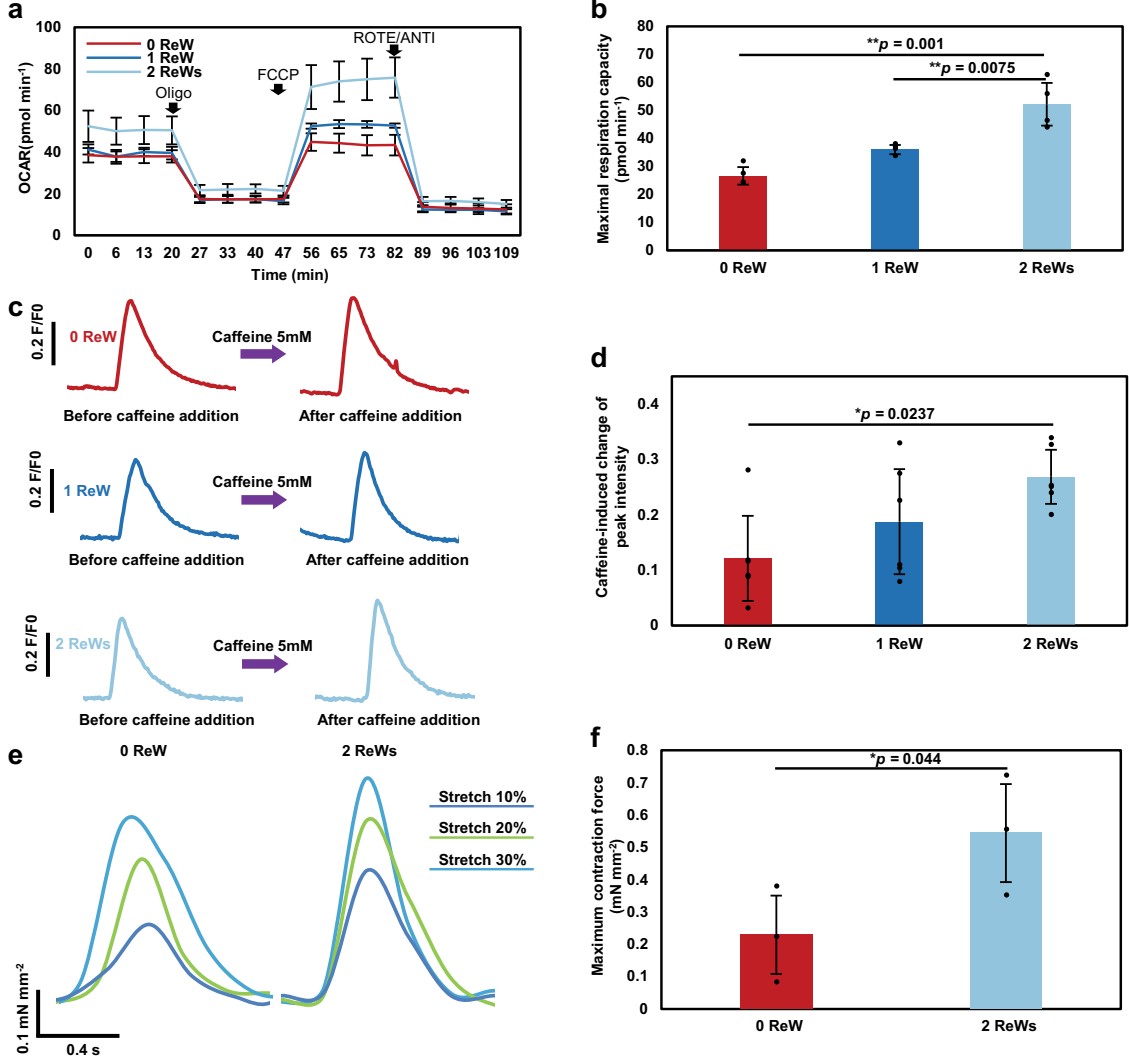

**Fig. 5 ReWs are associated with enhanced Ca$^{2+}$-handling properties. a** Results of Oxygen Consumption Rate (OCR) assays. Oligo, oligomycin; FCCP, carbonyl cyanide-4-(trifluoromethoxy) phenylhydrazone; ROTE/ANTI, rotenone and antimycin A. **b** Quantification of the maximal respiration capacity (mean ± s.d.; $n = 4$ biologically independent samples). $**P < 0.01$ (ANOVA). **c** Representative traces of Ca$^{2+}$ transient following administration of 5 mM caffeine to cardiomyocytes in SOTRs with zero ReW, one ReW, and two ReWs at day 7. SOTRs were immersed in room temperature medium to stop ReWs on day 6. **d** Caffeine-induced change in peak intensity in different groups (mean ± s.d.; $n = 6$ biologically independent samples). $*P < 0.05$ (ANOVA). **e** Force traces of SOTRs during progressive stretching. **f** Maximum contractile force of SOTRs (mean ± s.d.; $n = 3$ biologically independent samples). $*P < 0.05$ (one-tailed Student's $t$ test).

**Device manufacture**. PDMS (SYLGARD 184; Dow Corning, Midland, MI, USA) blocks were punched using a tissue puncher (Thermo Fisher Scientific, Waltham, MA, USA) to an inner diameter of 8 mm. After removal of the redundant central pillar, the remaining PDMS wells were collected, and PDMS wells and pillars with different diameters were centrally aligned and attached to the bottom of 24-well ultra-low attachment plates (Corning, Corning, NY, USA) as shown in Supplementary Fig. 1c. For micro-electrode array (MEA; Multi Channel Systems, Reutlingen, Germany) recording, the PDMS well and pillars were mounted to the MEA surface using silicone glue (Shin-Etsu Chemical, Tokyo, Japan) to fix the well and pillar to the bottom of the well. After drying and treatment with ultraviolet light for 30 min, the PDMS wells were ready for cardiomyocyte culture.

**SOTR generation**. Before cell seeding, single cardiomyocytes were filtered using a 40-μm cell strainer (BD Falcon; Becton Dickenson, Franklin Lakes, NJ, USA) and resuspended at a density of $2 \times 10^6$ cells mL$^{-1}$ in the culture medium containing 40% high glucose Dulbecco's modified Eagle's medium (DMEM) (Sigma-Aldrich), 40% IMDM (Iscove's modified Dulbecco's medium; Sigma-Aldrich), 20% fetal bovine serum (FBS; Gibco, USA), 1% minimum essential medium non-essential amino acid solution (Sigma-Aldrich), and 0.1% penicillin–streptomycin (Gibco), and 0.5% L-glutamine (Sigma-Aldrich). Then, $4 \times 10^5$ cells were plated in each PDMS culture well with a 3-mm pillar. For wells with other diameters, the cell

number was fixed to $4 \times 10^5$ or to a cell number that was proportional to the diameter. After plating, cardiomyocytes settled in the wells, aggregated, and congregated around the central pillar to form densely packed tissue rings within 2 days. The medium was changed to serum-free medium (culture medium without FBS) from day 2. After that, fresh medium was changed every 4 days. Before changing the medium, the fresh medium was preheated to 37 °C. The dishes were placed on top of a preheated metal block during medium change. The medium was pipetted gently and slowly into the wells. For dynamic culture, the dishes were mounted on a rotary shaker (NA-M301, Nissin, Japan), rotating at a speed of 26 r.p.m., in an incubator for 2 weeks. All the data in this study were collected under static culture condition for 14 days, unless specified otherwise.

**Optical mapping and measurement**. Progression of tissue assembly was observed using a fluorescence microscope (Olympus IX71; Olympus, Tokyo, Japan) equipped with a charge-coupled device (CCD) camera (Exiblue; Qimaging, Surrey, BC, Canada; or DP74; Olympus). GCaMP3 was excited from 450 to 490 nm, and fluorescence images of GCaMP3-positive cardiomyocytes were recorded with $8 \times 8$ binning of CCD pixels at 30 frames s$^{-1}$. Images were processed to obtain data using ImageJ (NIH, Bethesda, MD, USA) and MATLAB (R2014b; MathWorks, Natick, MA, USA) using a customized program. Activation-time mapping was performed using a custom plug-in in ImageJ, as previously described[57]. Briefly, a

Gaussian spatial convolution was applied with a 2 pixel radius to each frame. Subsequently, the time-series data were temporal low pass filtered for noise reduction, the upstrokes of cell activation were detected, and a linear activation map was calculated between subsequent activations independently for each pixel.

**Voltage-sensitive dye staining**. The voltage dye staining was performed according to the manufacturer's protocol (FluoVolt™, Thermo Fisher Scientific). Five microliters of FluoVolt™ dye was diluted in 50 μL PowerLoad and then suspended in 5 mL of high glucose DMEM (Sigma-Aldrich) preheated to 37 °C. Then, 200 μL of the dye solution was applied to the well containing SOTRs for 15 min. After removing the dye solution, the tissues were washed with DMEM twice.

**Transient and electrophysiological Ca$^{2+}$ characterization**. Ca$^{2+}$ transience was recorded by fluorescence imaging of GCaMP3-positive cardiomyocytes with the same experimental setup as that described for optical mapping and measurement. To characterize the spontaneous beating of SOTRs, ReWs were removed from the SOTRs 1 day before recording by emerging the SOTRs into the medium at room temperature for 1 min. The 5 mM caffeine (Wako Pure Chemical Industries Ltd. Osaka, Japan) was added directly to the chamber containing the SOTRs during imaging, as described in the figure legends.

Extracellular recording of field potentials (FPs) was performed using an MEA data-acquisition system (USB-ME64; Multi Channel Systems). Signals were recorded from days 5 and 6 after cardiomyocyte seeding, and data were collected and processed using MC_Rack (Multi Channel Systems) or LabChart (ADInstruments, Dunedin, New Zealand). The amplitude, QT interval, and beat rate were determined by analyzing the wave form of the FP.

**Histology**. SOTRs were rinsed three times with phosphate-buffered saline (PBS), fixed in 4% paraformaldehyde (PFA) in PBS for 30 min, and embedded in paraffin. Thin sections were sliced and stained with HE (Muto Chemical Corporation, Tokyo, Japan), and observation was carried out using a CKX41 microscope (Olympus).

**Immunostaining and imaging**. SOTRs were fixed in 4% PFA at room temperature for 30 min, permeabilized with 0.5% (v/v) Triton X-100 in Dulbecco's (D)-PBS at room temperature for 1 h, and incubated in blocking solution [0.1% (v/v) Tween-20, 5% (v/v) normal donkey serum, 3% (v/v) bovine serum albumin, and 5% (v/v) normal goat serum in D-PBS] at 4 °C for 16 h. SOTRs were then incubated with the primary antibodies anti-troponin T2 (TnT2; mouse monoclonal IgG; 1:200; SC-20025; Santa Cruz Biotechnology, Dallas, TX, USA), anti-α-actinin (mouse monoclonal IgG, 1:1000; A7811; Sigma-Aldrich), or anti-β-MHC (mouse MYH7 monoclonal IgM, 1:100; SC-53089; Santa Cruz Biotechnology) at 4 °C overnight. SOTRs were then rinsed with PBS and incubated with secondary antibodies diluted 1:300 in blocking buffer (Alexa Fluor 594 anti-mouse IgG; 715-586-150; and DyLight-594 anti-mouse IgM; 715-516-020; Jackson ImmnoResearch, West Grove, PA, USA) at room temperature for 1 h. 4′-6-Diamidino-2-phenylindole (DAPI; 300 nM; Wako Pure Chemical Industries, Ltd.) was used to counterstain nuclei at room temperature for 30 min, after which images were captured using a confocal microscope (FV1200; Olympus). cardiomyocyte orientation within SOTRs was determined using the Fourier component analysis plug-in "Directionality" in ImageJ[46] (NIH) that calculated the orientation distribution for the red color channel. Fluorescence quantification of β-MHC was performed by calculating the gray value averaged over the area in all sample groups, with all the fluorescence values were normalized to that of zero ReW group.

**Transmission electron microscopy (TEM)**. SOTRs were washed with 0.1 M Sorenson's buffer (pH 7.4) twice and fixed with 2.5% glutaraldehyde (Sigma-Aldrich) in 0.1 M Sorenson's buffer (pH 7.4), after which samples were post-fixed with 1% OsO$_4$ in Sorenson's buffer. The samples were then embedded, sliced, and stained with lead citrate and examined under a JEOL1010 transmission electron microscope (JEOL Ltd., Tokyo, Japan).

**Mitochondrial respiration assay**. Mitochondrial function was analyzed using a Seahorse XF96 extracellular flux analyzer (Agilent Technologies, Carlsbad, CA, USA). After a 14-day culture, SOTRs were dissociated into a single-cell suspension by stirring for 30 min in protease solution, followed by cell seeding onto a cell culture microplate (Agilent Technologies) at a density of 20,000 cells per XF96 well. OCR assays were performed 3 days after seeding, and the culture medium was changed to the base medium (Seahorse XF assay media supplemented with 1 mM sodium pyruvate; Life Technologies, Carlsbad, CA, USA). Substrates and inhibitors were injected during measurements at a final concentration of 3.5 μM 4-(tri-fluoromethoxy) phenylhydrazone (FCCP; Seahorse Bioscience, Billerica, MA, USA), 1 μM oligomycin, 0.5 μM antimycin, and 0.5 μM rotenone for the MitoStress assay.

**Flow cytometry**. Before being used for generating SOTRs, hiPSC-derived cardiomyocytes were fixed in 4% PFA at room temperature for 30 min, permeabilized with 0.5% v/v Triton X-100 in Dulbecco's (D)-PBS at room temperature for

30 min, incubated with anti-TnT2 antibodies (mouse monoclonal IgG, 1:200; Santa Cruz Biotechnology: SC-20025) or isotype-matched antibodies (BD Phosphoflow: 557782) at 37 °C for 30 min, washed with D-PBS, and then incubated with Alexa Fluor 488 anti-mouse IgG (1:500; Jackson ImmoResearch: 715-546-150). Cells were then washed twice with D-PBS and analyzed using a FACS Canto II flow cytometer (BD Biosciences, USA) and the FlowJo software (Treestar Inc., USA). Data shown are representative of four independent experiments.

**Contractility analysis**. SOTR contractility was measured using a micron-scale mechanical-testing system (MicroSquisher; CellScale Biomaterials Testing, Waterloo, ON, Canada). ReWs were stopped by exchanging the medium at room temperature 1 day before the measurement, and the SOTR was removed from the pillar and cut into a strip. Then, the strip was fixed on a stage and immersed in the culture medium warmed at 37 °C. A cantilever beam with a diameter of 0.30 mm was pressed onto the SOTR (Supplementary Fig. 14). The beam were lower to stretch the SOTR strip into different lengths. At each length, the beam was held for 50 s and the force was calculated by cantilever beam deflection in response to differential displacement.

**Quantitative polymerase chain reaction**. Total RNA in SOTRs was harvested using Trizol reagent (Life Technologies) according to the manufacturer's instructions, and RNA concentration was determined using a NanoDrop1000 spectrophotometer (Thermo Fisher Scientific). Complementary DNA (cDNA) was synthesized with a First-strand synthesis kit (TaKaRa, Shiga, Japan) and analyzed by qPCR using SYBR Green PCR master mix (Life Technologies) and the qBiomarker validation PCR array (IPHS-102A; Qiagen, Hilden, Germany) in a 96-well format according to the manufacturer's instructions. Cycling conditions were set as follows: initial denaturation at 95 °C for 10 min, followed by 40 cycles at 95 °C for 15 s and 60 °C for 70 s. Reactions were performed in a StepOnePlus real-time PCR system (Life Technologies). Gene expression was determined using the $2^{-\Delta\Delta Ct}$ method and relative to *glyceraldehyde 3-phosphate dehydrogenase* expression. Heatmaps were created by the "pheatmap" package (https://cran.r-project.org/web/packages/pheatmap/index.html), and clustering order was produced using the Ward.D clustering algorithm (https://stat.ethz.ch/R-manual/R-devel/library/stats/html/hclust.html).

**RNA quantification and qualification for RNA-sequencing**. RNA degradation and contamination was monitored on 1% agarose gels. RNA purity was determined using a NanoPhotometer spectrophotometer (Implen, Westlake Village, CA, USA). RNA concentration was measured using a Qubit RNA assay kit in a Qubit 2.0 fluorometer (Life Technologies). RNA integrity was assessed using the RNA Nano 6000 assay kit and the Agilent Bioanalyzer 2100 system (Agilent Technologies).

**Library preparation for RNA-sequencing**. A total of 1.5 μg RNA per sample was used as input material for RNA sample preparations. Adult normal heart RNA was purchased from BioChain (USA). Sequencing libraries were generated using a NEBNext UltraTM RNA library prep kit Illumina (New England Biolabs, Ipswich, MA, USA) according to the manufacturer's instructions, and index codes were added to attribute sequences to each sample. Briefly, mRNA was purified from total RNA using poly-T oligo-attached magnetic beads, and fragmentation was performed using divalent cations under an elevated temperature in NEBNext First-Strand Synthesis reaction buffer (5×; New England Biolabs). First-strand cDNA was synthesized using a random hexamer primer and M-MuLV reverse transcriptase (RNase H−; New England Biolabs). Second-strand cDNA synthesis was subsequently performed using DNA polymerase I and RNase H. Remaining overhangs were converted into blunt ends via exonuclease/polymerase activities. After adenylation of the 3′ ends of the DNA fragments, NEBNext Adaptor (New England Biolabs) with a hairpin loop structure was ligated to prepare for hybridization. To select cDNA fragments of the correct length, the library fragments were purified using the AMPure XP system (Beckman Coulter, Beverly, MA, USA), after which 3 μL USER enzyme (New England Biolabs) was incubated with size-selected, adaptor-ligated cDNA at 37 °C for 15 min, followed by 5 min at 95 °C. PCR was performed using a Phusion high-fidelity DNA polymerase, universal PCR primers, and an Index (X) primer (New England Biolabs). Products were purified (AMPure XP; Beckman Coulter), and library quality was assessed using the Agilent Bioanalyzer 2100 system (Agilent Technologies).

**Clustering and sequencing**. Clustering of the Index-coded samples was performed on a cBot cluster generation system using a HiSeq 4000 PE cluster kit (Illumia, San Diego, CA, USA) according to the manufacturer's instructions. After cluster generation, the library preparations were sequenced on an Illumina Hiseq 4000 platform (Illumina), and 150-bp paired-end reads were generated.

After quality control, the paired-end reads were aligned to the hg19 human reference genome (UCSC; https://genome.ucsc.edu/) using TopHat (v2.0.12; https://ccb.jhu.edu/software/tophat/index.shtml)[58]. We quantified the relative expression level as log$_2$(FPKM + 1) to perform PCA analysis and hierarchical clustering. Uniquely mapped reads counted by HTSeq[59] were used to conduct differential expression analysis in DEseq2[60]. A total of 349 genes ($P < 0.05$, fold change >1.5) were selected for the GO analysis. We plotted heatmaps using the "pheatmap" package in R (https://cran.r-project.org/web/packages/pheatmap/

index.html) and performed GO enrichment analysis using Metascape[61] (http://metascape.org/).

**Statistics and reproducibility**. All quantitative data are presented as the mean ± standard deviation of the mean (mean ± s.d.). The differences between experimental groups were analyzed by one-tailed Student's $t$ test (between two groups) or ANOVA (one-way analysis of variance), followed by Tukey's post hoc test (among three or more groups). A $P < 0.05$ was considered statistically significant.

**Reporting summary**. Further information on research design is available in the Nature Research Reporting Summary linked to this article.

## Data availability

The RNA-sequencing data in this paper could be visited under the accession number GEO: GSE140466. The other datasets generated during the current study are available from the corresponding author on reasonable request.

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

## Acknowledgements

Funding was provided by the Japan Society for the Promotion of Science (JSPS): Grant-in-Aid for Challenging Exploratory Research (B) (15K13911, to L.L.), Grants-in-Aid for Scientific Research (C) (19K12801, to J.L.), Grants-in-Aid for Young Scientists (B) (17K14624, to J.L.). Funding was also provided by the Japan Agency for Medical Research and Development (AMED). This work was also supported by the Ministry of Science and Technology of China (2015CB910300, to C.T. and 2018YFA0800504 to Y.Z.).

## Author contributions

J.L., L.L., C.T., and Y.S. conceived the project, and J.L., L.Y., I.M., C.Y., L.L., C.T., S.M., and Y.S., designed the experiments. J.L. and J.Q. fabricated the device. Y.Z. and J.L. performed electron microscopy and analyzed the images. Y.S. generated the GCaMP3-expressing hiPSC line. I.M. performed hiPSC culture and cardiomyocyte differentiation. J.L., L.Y., X.Q., Y.H., M.H., and N.F. conducted the immunostaining experiments. L.Y., J.D., Y.Z., and F.T. performed the qRT-PCR and RNA-sequencing experiments. J.L., M.H., and J.Q. performed the electrophysiological characterization. J.L., I.M., and J.L. performed flow cytometry. L.Z. and C.T. performed the simulation experiments. All authors contributed to data analysis and interpretation. J.L., L.Z., Y.S., C.T., and L.L. wrote the manuscript.

## Competing interests

The authors declare no competing non-financial interests but the following competing financial interests: L.L. and J.L. filed a provisional Japanese patent application (2016-255258) based on the research presented here. I.M. is a shareholder of Myoridge Co. Ltd. Y.S.'s laboratory received funding from the TERUMO Company.
