## [Peer Review File · Communications Biology]

Reviewers' comments:

Reviewer #1 (Remarks to the Author):

The work presented by Liu et al., introduces a bioengineering method to create ring-shape cardiac tissue constructs, made of hiPSC-CMs, and further maturation of these tissues using electrical pacing. While the work has a cohesive and well-structured form, there still seems to be a number of questions and issues that need to be addressed in order to improve the significance and impact of the work to the level that merits publication in Nature Communications Biology.

There are already a good number of published works on engineered cardiac tissue rings and their enhanced function/maturation (e.g. works by Zimmermann, Eschenhagen, Sato, Jerry Yang, and others). Further, there are multiple works on the effect of electrical pacing on enhanced maturation of CMs (e.g., doi: 10.1038/s41586-018-0016-3 and doi: 10.1007/s12265-013-9510-z). Extensive in vitro and in vivo studies have been already performed on these ring-shape cardiac tissues. So, the work presented here would need to provide sufficient amount of additional insight into the field. Here are some suggestions to add further impact/novelty to the work:

- . Testing dynamic culture versus static.
- . Long-term culture effects: seems like the longest these tissues were cultured here was 2 weeks (this is not quite clear in Methods section). How would the results change after several months of culture (with/without pacing)?
- . Incorporation of other cardiac cells?
- . Comparing a variety of shapes/geometries (see my comment below)

Below are some more specific (some minor) comments:

- The logic for choosing circular (ring) geometry for the engineered cardiac tissues: would be necessary to present some justification/discussion on why this specific geometry was used. What would potential implications/applications of such tissue shape, and how that compares with other forms (e.g., a rectangular or sheet-like construct).
- Control studies: following up on comment above, it would be very helpful to have results from another tissue shape/structure as control. Just a monolayer, conventional CM sheet could serve the purpose. Have authors looked into the effect of 0, 1, and 2 TW on CMs in monolayer sheets? One would wonder whether this ring shape induces any additional benefit or the observed maturation evidence could be simply generated by the pacing regimens.
- In Figure 1: Graphs in panels c, f, and g: missing statistics and P values. All graphs in the study must have n and P values, reported in the figure and/or in its caption.
- Figure 2: needs better labeling and structuring the results. In panel (a): each row needs to be labelled properly. Is the top row simulation and the bottom row experimental results? This is not clearly explained in the caption. Also, stats are missing in the graphs (P value).
- Figure 3: Panel (f) is missing a " β -MHC" label on top. Also, it is not clear what technique was used for quantifications in (g)? were they obtained from confocal imaging, qPCR, etc.? This needs to be clarified both in the figure and in the text in Results section.
- Figure 4: I recommend adding insets with zoomed-in views of α -actinin and cTnT staining, demonstrating evidence of maturation - to compare sarcomeric organization in those confocal images. Current low-mag images do not clearly show those features.

- Figure 5 - contractility analysis: It is not clear for me how the authors quantified the contractile force. In Methods, they briefly mention: "A cantilever beam with a diameter of 0.30 mm was pressed onto the SOTR...". This needs further elaboration. A figure (schematic or actual photo of the set-up) would be very helpful for readers to understand the approach. The precision and reproducibility of this approach shall be discussed.

- In Suppl Figure 2: Graphs presented in panel (c) are missing error bars / statistics. All graphs in the study must have n and P values, reported in the figure or in its caption.

Reviewer #2 (Remarks to the Author):

The study by Li et al. describes a self-organizing cardiomyocyte culture model. Starting from a monolayer, human iPSC-derived cardiomyocytes aggregate into a ring structure around a central pillar. Similar observations have been reported by others (e.g., more recently by Goldfracht et al. 2019 using a matrix supported model). The use of genetically encoded calcium sensors, GCaMP, allowed for the discrimination of cardiomyocyte rings with reentry type phenomena, designated as traveling waves (TWs). The authors performed a number of molecular, morphological, and functional analyses to determine the state of the cardiomyocyte rings (SORTs) without reentry and with simple and more complex reentry phenomena. In addition, they created a mathematic model to simulate the experimental data. Taken together, the model is not new and the observation somewhat anticipated. The interpretation that the TWs supported maturation is not unequivocally supported by data. It seems that differences in SORT organization may have resulted in the observed phenomena. The experiments are performed well and the speculated use in drug testing is valid. The mathematical modelling is an interesting addition to the study. I have a couple of suggestions to strengthen the study.

Specific suggestions:

The authors should restrict themselves to describe the model and the specifically observed phenomena. The model itself may be of use for drug screening, for example for compounds that interfere with conduction. All speculations as to the use in regeneration, use of other cell types, or use in maturation should be deleted. There is simply no evidence reported in support of these claims.

There is no evidence for a mechanistic role of the TWs in maturation. The TWs are a simple phenomenon observed by the authors. The association of TWs with better structure and molecular markers does not support the claim for a cause or a relation to the cause. It seems to be simply a random presentation of different phenotypes, i.e., high and lower degree of organization. The challenge is really to control and make use of these phenomena.

Related to the previous statement, the differently observed electrophysiological phenomena appear to form at random. It is difficult to understand how the authors would aim for the creating of SORTs with no or multiple reentry phenomena.

The observation of SORTs with TW and without could be due to the presence of pacemaker cells in the SORTs without TW. An important experimental addition to the study would be to assess whether the SORTs without TW could be transformed into SORTs with TW by external electrical stimulation. From an electrophysiological point of view TWs could be interpreted as sign of immaturity. The slow conduction velocity supports his argument.

Cell content and composition must be evaluated at the time point of electrophysiological analyses.

The data on force measurement must be rephrased or deleted. The microsquisher uses tissue compression to record tissue expansion. This is only indirectly related to force of contraction, which is measured in isometrically suspended muscle. In that case, force can be reported in mN/mm² muscle cross section. Using the microsquisher different type of data is collected, which must not be reported as mN/mm² unless the area is the area of compression. In this case, a clarification of the differences in force metric must be presented and discussed.

Minor:

Title: must be changed; there is no evidence for TW induced maturation

Line 43: the claim to report a new idea to mature cardiomyocytes is not warranted

Line 75: there is no rapid pacing by spiral waves, this terminology is restricted to specific electrophysiological phenomena. The authors describe reentry phenomena of different types.

Line 100: the claim that thick tissue has formed is at least debatable. I recommend to insert data on the mean \pm SD or SEM thickness for clarity.

Line 254: the term ring organoid is misleading. An organoid cannot be composed of only cardiomyocytes. The authors are describing a classical aggregation culture. Which is in this case controlled by the central pillar. Similar models have been around for many years.

Lines 284/285: must be delete or rephrased; there is no evidence of frequency depended maturation; the TWs are very likely a phenomenon of better cardiomyocyte arrangement, as demonstrated by the FF study.

Li et al. *“Rapid pacing by circulating traveling waves improves maturation of hiPSC-derived cardiomyocytes in self-organized tissue ring”* Manuscript no. COMMSBIO-19-0780-T

Responses to the comments from the reviewers:

•**Response:** We are very grateful to the reviewers for their comments and suggestions. Accordingly, we made a number of modifications and improved our manuscript. Below, we describe the modifications and provide our responses to reviewers' comments (blue text).

Reviewers' comments:

Reviewer #1 (Remarks to the Author):

The work presented by Liu et al., introduces a bioengineering method to create ring-shape cardiac tissue constructs, made of hiPSC-CMs, and further maturation of these tissues using electrical pacing. While the work has a cohesive and well-structured form, there still seems to be a number of questions and issues that need to be addressed in order to improve the significance and impact of the work to the level that merits publication in Nature Communications Biology.

There are already a good number of published works on engineered cardiac tissue rings and their enhanced function/maturation (e.g. works by

Zimmermann, Eschenhagen, Sato, Jerry Yang, and others. Further, there are multiple works on the effect of electrical pacing on enhanced maturation of CMs (e.g., doi: 10.1038/s41586-018-0016-3 and doi: 10.1007/s12265-013-9510-z). Extensive in vitro and in vivo studies have been already performed on these ring-shape cardiac tissues. So, the work presented here would need to provide sufficient amount of additional insight into the field.

•**Response:** We thank the reviewer for the comments and suggestions. We agree with the referee that the tissue ring is not a new tissue type, and there are indeed reports on engineering cardiac tissue rings and promoting their maturation. Despite of these findings, we would like to mention the following:

1. In the best of our knowledge, this is the first study proving that the human cardiomyocytes could spontaneously and stably beat at high frequency (~ 4 Hz) for weeks.
2. We found and describe here, for the first time, the connection between the rapid pacing by spontaneous traveling waves and the maturation of cardiomyocytes. Thus, this technology could be used as a supplement to electrical pacing, which has been proved to be important for the cardiomyocytes maturation [1, 2].
3. Compared to the conventional electrical pacing method, the traveling wave based maturing technology does not need external setup. Users could simply modify the conventional petri dish to obtain matured cardiomyocytes.

In order to better compare with our technology to previous reports, we modified the introduction and discussion part and added more data (shown in following sections) and references suggested by reviewer.

Here are some suggestions to add further impact/novelty to the work: Testing dynamic culture versus static.

According to this suggestion, we cultured the ring on a rotary shaker (NAM301, Nissin, Japan) at 26 rpm for 2 weeks. During this period, the beating frequency of CMs in the ring with 2 TWs (Supplementary Fig. 7) significantly increased from 3.30 ± 0.39 Hz to 3.89 ± 0.18 Hz at day 6, and from 3.92 ± 0.69 Hz to 5.57 ± 0.06 Hz at day 14. On another hand, there were no significant differences between the 0 TW groups with or without rotatory culture (dynamic culture). These data indicate that the rotatory culture could improve the availability of oxygen and glucose to cardiomyocytes [3] in the TW group and thus increase the beating frequency. Despite the improvement, we also found that the TW occurrence ratio in the dynamic group dropped significantly to 50% on day 6 and 40% on day 14, which could be possibly a result of the disturbance caused by the dynamic medium flow. Currently, we are still working on improve the well design, the future design with reduced medium disturbance could help further improve the yield of the matured tissue ring. Accordingly, the manuscript has been modified.

b

Day 6		Day 14	
Static	Dynamic	Static	Dynamic
85.79%	50%	71.51%	40%

Supplementary Figure 7. The comparison between dynamic and static culture of SOTRs. (a) Beat rates of SOTRs at different culture times, (mean \pm s.d.; 2 TWs-Static: n = 10; 2 TWs-Dynamic: n = 8; 0 TWs-Static: n = 8; 0 TWs-Dynamic: n = 3.) $**P < 0.01$; $***P < 0.001$. (b) TW occurrence ratio for static culture and dynamic culture, Static: n = 204. Dynamic: n = 20.

. Long-term culture effects: seems like the longest these tissues were cultured here was 2 weeks (this is not quite clear in Methods section). How would the results change after several months of culture (with/without pacing)?

•**Response:** We apologize for the unclear description in the Methods section. A clearer description has been added to the method section.

According to the reviewer's suggestions, we cultured the rings in the presence of TWs for more than 3 months (Supplementary Fig. 6a of the revised manuscript). During long culture, the beating frequency of the one TW group did not increase with time (Supplementary Fig. 6b) and the sarcomere length of this group at day 100 did not increase further compared to the observations at day 14 (Supplementary Fig. 6c, d). This indicated that the rapid pacing maturation process could be limited within a specific time window, similar to the findings of a previous report [1].

Supplementary Figure 6. Maintenance of TWs in SOTRs for >89 days. (a) Fluorescence images of a SOTR with one TW at different culture times. (b)

The beat rate (Hz) of SOTRs with one TW during a long-term culture (mean \pm s.d.; n = 3). (c) Sarcomere lengths of CMs within SOTRs on day 14 and day 100 [mean \pm s.d.; Day 14 samples: 0 TWs (n = 5); 1 TW (n = 5); 2 TWs (n = 4); Day 100 samples: 0 TWs (n = 4); 1 TW (n = 3)]. * P < 0.05; ** P < 0.01.

Accordingly, we have added the following sentences to the Discussion section:

Line 318: “The beating frequency and sarcomere length of TW trained CMs did not increase with culture time (>89 day) longer than 14 days (Supplementary Fig. 6). This indicated that the rapid pacing maturation process could be limited within a specific time window, similar to the findings of a previous report [1].”

In Methods section:

Line 372: “All the data in this study was collected under static culture condition for 14 days, unless specified otherwise”.

. Incorporation of other cardiac cells?

•**Response:**

We have evaluated the relative ventricular to atrial CM composition in the differentiated CMs in our previous report by FACS and patch clamp [4]. The results indicated that the iPS cells (IMR90-1) derived CMs obtained by our differentiation protocol are composed of mostly ventricular cells (60% for MLC2v-positive and 10% for MLC2v/MLC2a-positive) and few atrial cells (MLC2a) and pace maker cells. As for the 253G1 cells (cell line used in this

study) derived CMs, the cell composition is similar. (70% for MLC2v-positive cells, 10% for MLC2v/MLC2a-positive cells, 0.2~0.3% for MLC2a-positive cells, 10% for pace maker cells, Response Figure 3). We would like to mention that since the data for 253G1 cells is similar to our previous report [4], we would like to include the data in Response Figure 3 in another of project that compares the impact of culture methods (Suspension and Attachment) on differentiation efficiency.

. Comparing a variety of shapes/geometries (see my comment below)

Below are some more specific (some minor) comments:

- The logic for choosing circular (ring) geometry for the engineered cardiac tissues: would be necessary to present some justification/discussion on why this specific geometry was used. What would potential implications/applications of such tissue shape, and how that compares with other forms (e.g., a rectangular or sheet-like construct).

•**Response:** We have chosen the ring because it could offer the closed-loop circuit for the lasting TW, which could pace and mature the CMs without using any external experimental setup. Moreover, the matured tissue ring could be easily used for other applications, such as previous reported drug assessment [5] and transplantation [6]. Secondly, we have also tried to repeat the TW phenomenon in a number of geometries of engineered cardiac tissues: 4-point star formation, 5-point star formation and 2D sheet-like construct.

The ring formed in the 4-point star form is similar to the rectangular one (Supplementary Fig. 2). We found that although the ring could be formed on the 4-point pillar, and TWs could be found in most of the rings. However, the tissue was not homogeneous in different areas of the ring: the corner area tissue is thinner than that in the edge area. These unequally distributed tissues could break easily during culture or in future applications. When we improved the 4-point star geometry to 5-point star one, the tissue becomes more homogeneous and similar to the circular condition. Compared with the star formation, the circular ring was more equally and stably organized while maintaining the TWs with a higher ratio (Supplementary Fig. 2b). Thus, we chose the circular ring for further experiments.

This information has been introduced in the Results section:

Line 101: “Besides circular geometry, other geometries have also been tested, such as 4-point star, 5-poin star (Supplementary Fig. 2a), and the geometry between rectangular and circular could also be obtained. However, the formation of the tissues throughout different areas of these templates was less homogeneous than that of the tissues in the circular geometry. In addition, the circular ring tissues were more equally and stably organized while maintaining the TWs with a higher ratio (Supplementary Fig. 2b). Thus, we chose the circular ring to engineer cardiac tissues.”

b

Supplementary Figure 2. hiPSC-CMs in different templates. (a) The rings are formed within 6 days. The red and blue arrows indicate thick and thin area in tissue ring, respectively. (b) The percentage of occurrence for TWs in different templates on day 6 (4-point star: n = 30; 5-point star: n = 6; Circular: n = 204)

-Control studies: following up on comment above, it would be very helpful to have results from another tissue shape/structure as control. Just a monolayer, conventional CM sheet could serve the purpose. Have authors looked into the effect of 0, 1, and 2 TW on CMs in monolayer sheets? One would wonder whether this ring shape induces any additional benefit or the observed maturation evidence could be simply generated by the pacing regimens.

•**Response:** Actually, we are also preparing another work on using TW to promote the maturation of 2D tissue sheets. We have confirmed that the TW in 2D tissue sheets could also lead to CM maturation, such as improved gene expression and conduction speed, within two weeks. (Response Figure 1 and 2)

We could conclude the differences between 3D ring shape and monolayer form:

1. Alignment of CMs, an important feature in adult heart, could not be observed in the 2D tissue.
2. The CMs in the 3D tissue does not require substrate while 2D tissue needs to adhere on substrate, which could allow larger exercise activity in 3D tissue and thus lead to higher maturation.
3. The 3D ring could be readily used for transplantation, such as previous reported transplantation of EHT (with no TW maturation) [6, 7], the 2D tissue would have to be dissociated and replated on other scaffold. However, the dissociation process could affect the original state of CMs such as cell-cell connection and organization.

Since the 2D TW could be out of the scope of present study and we are initiating a new research project on 2D tissue sheets for drug screening application, we thus consider to put the preliminary 2D data into the new study.

- In Figure 1: Graphs in panels c, f, and g: missing statistics and P values. All graphs in the study must have n and P values, reported in the figure and/or in its caption.

•**Response:** The P value and the n number have been added to the figure and the caption.

Figure 1. TWs promoted rapid beating of cells in SOTRs. (a) Schematic describing SOTR formation. (b) Bright-field images of hiPSC-CMs in the template. The red arrows indicate the edge of cardiac tissues in the template, and the dashed lines indicated the PDMS block and pillar boundaries, respectively. Pillar diameter = 3mm. (c) Quantification of the width of the SOTRs on the indicated culture day (mean \pm s.d.; $n = 4$). $**P < 0.01$. (d) Activation map of GCaMP3-positive SOTRs with zero, one, or two TWs. The red arrows indicate the propagation direction of the action potential. (e) GCaMP3-fluorescence signal at a fixed position on the ring of SOTRs with

zero, one, or two TWs on day 6. (f) Beat rates of SOTRs at different culture times (mean \pm s.d.; 2 TWs: n = 10; 1 TW: n = 12; 0 TW: n = 8). * $P < 0.05$; ** $P < 0.01$; (g) The wave speed of spontaneous beating in SOTRs with zero, one, or two TWs after 14 days of culture; for TW groups, the speed was recorded after the TWs are stopped and the spontaneous beating was recovered. (mean \pm s.d.; 2 TWs: n = 4; 1 TW: n = 4; 0 TW: n = 6). * $P < 0.05$. (h) The percentage of occurrence for different numbers of TWs on days 6 (n = 204) and 14 (n = 186), respectively.

-Figure 2: needs better labeling and structuring the results. In panel (a): each row needs to be labelled properly. Is the top row simulation and the bottom row experimental results? This is not clearly explained in the caption. Also, stats are missing in the graphs (P value).

•**Response:** We apologize for the confusion. Both panels are from the simulation results. The top row is membrane voltage of the ring CMs with different TWs. The bottom row is activation map for the ring CMs with different TWs. The figure has been updated for clarification.

Figure 2. Accurate reproduction of TW features in SOTRs using a mathematical model. (a) Examples of simulation results of the model (upper panel) and the activation map (lower panel). For the group with zero TWs, an action potential initiates from the left position of the ring and propagates in the opposite directions along the ring, after which the two waves meet and annihilate on another. The process will start again with new waves initiating at the same position periodically. For groups with one, two, or three TWs, stable TWs travel around the ring. Membrane potential and activation time are color coded. (b) The percentage of stable TWs in SOTRs at day 6 (diameter = 3 mm; n = 60 in the simulation and n = 204 in the experiment). (c) Beat rate

(Hz) of CMs in SOTRs with one TW at day 6 for different pillar diameters [experiment: 1-mm SOTR (n = 4); 3-mm SOTR (n = 12); 5-mm SOTR (n = 10)] $**P < 0.001$. (d) Similar information shown in (c) but with TW speed. (e) Speeds of one, two, or three TWs in SOTRs with a pillar diameter of 3 mm [experiment: mean \pm s.d.; 3 TWs (n = 4); 2 TWs (n = 10); 1 TW (n = 12)] $*P < 0.05$; $**P < 0.001$. (f) The maximum number of TWs in SOTRs with different pillar diameters.

- Figure 3: Panel (f) is missing a " β -MHC" label on top. Also, it is not clear what technique was used for quantifications in (g)? were they obtained from confocal imaging, qPCR, etc.? This needs to be clarified both in the figure and in the text in Results section.

•**Response:** The label has been updated in Figure 3, and the analysis technique has been clarified in the figure legend.

Line586: Quantification of β -MHC levels in SOTRs on day 14. The total integrated fluorescence of β -MHC retrieved from confocal imaging was normalized to that of SOTRs with zero TWs.

- Figure 4: I recommend adding insets with zoomed-in views of a-actinin and cTnT staining, demonstrating evidence of maturation - to compare sarcomeric organization in those confocal images. Current low-mag images do not clearly show those features.

•**Response:** The figure has been updated. Zoomed-in views have been added for both a-actinin and cTnT staining results.

Figure 4. TWs promote CM alignment. (a) Representative confocal images of SOTRs with zero, one, or two TWs. CMs were stained with anti-TnT2 (red) and DAPI (blue), revealing CM alignment in TW samples at day 14. Arrows mark the ring orientation. (b) Angular graph based on Fourier component analysis of TnT2 orientation distribution. (c) Representative confocal images of stained SOTRs with different TWs (red: α -actinin; blue: DAPI). Arrows mark the ring orientation at day 14, and the area delimited by a green dashed square is zoomed-in in each confocal image.

- Figure 5 - contractility analysis: It is not clear for me how the authors quantified the contractile force. In Methods, they briefly mention: "A cantilever beam with a diameter of 0.30 mm was pressed onto the SOTR...". This needs

further elaboration. A figure (schematic or actual photo of the set-up) would be very helpful for readers to understand the approach. The precision and reproducibility of this approach shall be discussed.

•**Response:** A detailed description of the force analysis has been added in the Methods section, and one additional figure has been added to clarify this method.

Supplementary Figure. 14 The contractility force recording set up.

Line463:

“**Contractility analysis**

SOTR contractility was measured using a micron-scale mechanical-testing system (MicroSquisher; CellScale Biomaterials Testing, Waterloo, ON, Canada). TWs were stopped by exchanging the medium at room temperature 1 day before the measurement, and the SOTR was removed from the pillar and cut into a strip. Then, the strip was fixed on a stage and immersed in culture medium warmed at 37°C. A cantilever beam with a diameter of 0.30 mm was pressed onto the SOTR (Supplementary Fig. 14). The beam were lower to stretch the SOTR strip into different lengths. At each length, the beam was held for 50 s and the force was calculated by cantilever beam deflection in response to differential displacement.”

- In Suppl Figure 2: Graphs presented in panel (c) are missing error bars / statistics. All graphs in the study must have n and P values, reported in the figure or in its caption.

•**Response:** The figure have been updated. Data from more samples were added to the figure. The corresponding n and P values are also added.

Supplementary Figure 5. Field potential recording by micro-electrode arrays. (a) Representative setup for FP recording by using electrodes underneath a SOTR on day 6. See also supplementary video 5. (b) Representative FP recorded by an electrode placed underneath a SOTR with or without TWs. (c) Frequency, QT interval (correlated with AP duration and

refractory period), and wave speed in SOTRs with or without TWs. [mean \pm s.d.; 0 TWs (n = 4); 1 TW (n = 4); 2 TWs (n = 3)]. ** $P < 0.01$.

Reviewer #2 (Remarks to the Author):

The study by Li et al. describes a self-organizing cardiomyocyte culture model. Starting from a monolayer, human iPSC-derived cardiomyocytes aggregate into a ring structure around a central pillar. Similar observations have been reported by others (e.g., more recently by Goldfracht et al. 2019 using a matrix supported model). The use of genetically encoded calcium sensors, GCaMP, allowed for the discrimination of cardiomyocyte rings with reentry type phenomena, designated as traveling waves (TWs). The authors performed a number of molecular, morphological, and functional analyses to determine the state of the cardiomyocyte rings (SOTRs) without reentry and with simple and more complex reentry phenomena. In addition, they created a mathematic model to simulate the experimental data. Taken together, the model is not new and the observation somewhat anticipated. The interpretation that the TWs supported maturation is not unequivocally supported by data. It seems that differences in SORT organization may have resulted in the observed phenomena. The experiments are performed well and the speculated use in drug testing is valid. The mathematical modelling is an interesting addition to the study. I have a couple of suggestions to strengthen the study.

Specific suggestions:

The authors should restrict themselves to describe the model and the specifically observed phenomena. The model itself may be of use for drug screening, for example for compounds that interfere with conduction. All speculations as to the use in regeneration, use of other cell types, or use in maturation should be deleted. There is simply no evidence reported in support of these claims.

•**Response:** We thank the reviewer for the comments and suggestions. We have carefully rephrase the sentences regarding the possible use of this model.

In the abstract part:

Line43: “The TW could also potentially be used for pacing the electrical excitable cells such as neuron and retina cells for various applications”

As for the application in maturation, we have added new data in the revised manuscript to prove that the maturation is caused by TWs, but not by other factors such as random presentation of different phenotypes.

There is no evidence for a mechanistic role of the TWs in maturation. The TWs are a simple phenomenon observed by the authors. The association of TWs with better structure and molecular markers does not support the claim for a cause or a relation to the cause. It seems to be simply a random

presentation of different phenotypes, i.e., high and lower degree of organization. The challenge is really to control and make use of these phenomena.

•**Response:** We thank the reviewer for the comments and suggestions. We have added the data of the ring with different TWs at day 2 after plating (Supplementary Figure 4). At day 2, the formation of ring and their TW number could just become stabilized (3D ring formation compared with initial monolayer formation). As shown by the staining, all samples, with or without TWs, all demonstrated randomly distributed, and poorly organized CMs and similar maturation. After two weeks culture, the rings with TW would demonstrate aligned structure and higher expressed molecular markers than those without TWs (Figure 4).

On another hand, it has been previously reported that the rapid electrical pacing would lead to CM maturation [1, 2]. The spontaneously originated TW in our study, although slower than that in those reports (~4 Hz vs. ~6 Hz), could still promote higher maturation than those without pacing.

In addition, the simulation model data indicated that the initial state of CMs within the SOTR leads to different TW number in the ring (Figure 2).

Taken together, these finding indicate that the maturation improvement could be the result of TW training, rather than vice versa. And we considered changing the title to:

“Circulating traveling waves rapidly pace and mature hiPSC-derived cardiomyocytes in self-organized tissue ring.”

Additionally, the following information has been added to the Result section:

Line 232:

“Moreover, on day 2 the SOTRs with or without TWs all demonstrated randomly distributed, and poorly organized CMs and similar maturation level (Supplementary Fig. 4). And the TW could also be induced in zero TW SOTR by rapid electrical stimulation at day 14 (Supplementary Video 6). Collectively, these data, together with the previous day 14 results, indicate that that the TW pacing resulted in improved maturation, rather than that the more mature ring could have more TWs.”

We agree with the reviewer that so far, it's still challenging to control the TW. Although we could use rapid electrical stimulation to induce TW in tissue ring that originally has no TW (Supplementary Video 6), but it is still difficult to generate more TW (two or more TWs) by the electrical stimulation. More future efforts are needed to improve the precise control for stimulation time and location.

Supplementary Figure 4. Representative confocal images of SOTRs with zero, one, or two TWs at day 2. CMs were stained with anti- α -actinin (red); anti- β -MHC (red); anti-TnT2 (green); anti-Vimentin (red) and DAPI (blue).

Related to the previous statement, the differently observed electrophysiological phenomena appear to form at random. It is difficult to understand how the authors would aim for the creating of SORTs with no or

multiple re-entry phenomena.

The observation of SORTs with TW and without could be due to the presence of pacemaker cells in the SORTs without TW. An important experimental addition to the study would be to assess whether the SORTs without TW could be transformed into SORTs with TW by external electrical stimulation. From an electrophysiological point of view TWs could be interpreted as sign of immaturity. The slow conduction velocity supports his argument.

•**Response:** We agree with the reviewer that the TWs appear at random and so far it is difficult to accurately control the appearance of TWs. Nonetheless, we do have tried to pace the SOTRs without TW (0 TW) at day 14 with a rapid electrical stimulus (Supplementary Video 6). The TW could be induced and maintained after the rapid pacing stopped.

Although the wave speed in the TW group is slower than that in the zero TW group, it could be caused by the shortened pacing interval and refractory period. [8-12] In addition, in our revised data (Figure 1g), we found that, after the TW was stopped and the SOTR begun to beat spontaneously, the conduction velocity of the two TWs group was considerably higher than that of the zero TW group, indicating improved maturation. This can be also confirmed by the upregulated expression of gap junction gene CX43/GJA1 in two TWs group (Supplementary figure 11).

Figure 1g. The wave speed of spontaneous beating in SOTRs with zero, one, or two TWs, for TW groups, the speed is recorded after TWs are stopped and the spontaneous beating is recovered. (mean \pm s.d.; 2 TWs: n = 4; 1 TW: n = 4; 0 TW: n = 6). * $P < 0.05$.

Supplementary Figure 11. FPKM values from RNA-sequencing data. The expression of *GJA1*, *GJC1*, *GJC1*, *CAV3*, *ADRB1* and *ADRB2* in different sample groups, which included adult heart, zero TW, one TW and two TWs samples.

The following description has been included in the Results section:

Line 115: “Notably, the wave speed in the TW groups was much lower (~ 2 cm/s) than that of the spontaneous beating group (0 TW; 5.93 ± 1.50 cm/s). It is possible that the slower speed together with the shortened pacing interval and refractory period might have been caused by the higher beating frequency in the TW groups relative to that in the zero TW group (Supplementary Fig. 5), which agrees with previous reports associated with excitable media [8-12]. Additionally, after the TW was stopped, the speed of

spontaneous beating in the two TW group increased significantly to more than 10 cm s^{-1} (Fig. 1g)”

Cell content and composition must be evaluated at the time point of electrophysiological analyses.

•**Response:** We have evaluated the relative ventricular to atrial CM composition in the differentiated CMs in our previous report by FACS and patch clamp [4]. The results indicated that the iPS cells (IMR90-1) derived CMs obtained by the differentiation protocol are composed of mostly ventricular cells (60% for MLC2v-positive and 10% for MLC2v/MLC2a-positive) and few atrial cells (MLC2a) and pace maker cells. As for the 253G1 cells (cell line used in this study) derived CMs, the cell composition is similar. (70% for MLC2v-positive cells, 10% for MLC2v/MLC2a-positive cells, 0.2~0.3% for MLC2a-positive cells, 10% for pace maker cells, Response Figure 3). In addition, we also used voltage dye to check the type of cells within the SOTRs, and the action potential curve confirmed that the CMs within the SOTR ring are mostly ventricular cardiomyocytes (Supplemental Figure 3).

[REDACTED]

[REDACTED]

[REDACTED]

[REDACTED]

[REDACTED]

[REDACTED]

[REDACTED]

[REDACTED]

[REDACTED]

Supplemental Figure 3. Representative fluorescence images of SOTRs with zero, one, or two TWs at day 6. CMs were stained with FluoVolt voltage dye according to the manufacturer’s protocol. The dashed line marks the tissue area in the images. The numbered squares mark the area where the membrane potential was recorded.

The data on force measurement must be rephrased or deleted. The microsquisher uses tissue compression to record tissue expansion. This is only indirectly related to force of contraction, which is measured in isometrically suspended muscle. In that case, force can be reported in mN/mm² muscle cross section. Using the microsquisher different type of data is collected, which must not be reported as mN/mm² unless the area is the

area of compression. In this case, a clarification of the differences in force metric must be presented and discussed.

Response. We apologize for the unclear description of the force signal recording; the updated description of the measurement setup has been added to the manuscript and it is also described below. Using the microsquisher, the force is in fact recorded by stretching (Supplementary Fig. 14), rather than by compression.

Supplementary Figure. 14 The contractility force recording set up.

Line 463:

“Contractility analysis

SOTR contractility was measured using a micron-scale mechanical-testing system (MicroSquisher; CellScale Biomaterials Testing, Waterloo, ON, Canada). TWs were stopped by exchanging the medium at room temperature 1 day before the measurement, and the SOTR was removed from the pillar and cut into a strip. Then, the strip was fixed on a stage and immersed in culture medium warmed at 37°C. A cantilever beam with a diameter of 0.30 mm was pressed onto the SOTR (Supplementary Fig. 14). The beam were lower to stretch the SOTR strip into different lengths. At each length, the

beam was held for 50 s and the force was calculated by cantilever beam deflection in response to differential displacement.”

Minor:

Title: must be changed; there is no evidence for TW induced maturation

•**Response:** As previously indicated, we have updated the title to “Circulating traveling waves rapidly pace and mature hiPSC-derived cardiomyocytes in self-organized tissue ring.”

Line 43: the claim to report a new idea to mature cardiomyocytes is not warranted

•**Response:** The sentence has been modified to:

Line 43: “The TW could also potentially be used for pacing the electrical excitable cells such as neuron and retina cells for various applications.”

Line 75: there is no rapid pacing by spiral waves, this terminology is restricted to specific electrophysiological phenomena. The authors describe reentry phenomena of different types.

•**Response:** We would like to mention that the pacing could not only referred to electrophysiological phenomena, but also to the athletic technique [13], which all refer to artificially regulation of the heart rate and movement [14]. Moreover, we also could find the words such as “Motor-paced racing”, “Pacing by car” and “Pacing by motorcycle”, which all referred to the modulating of

movement by other objects. We thus consider pacing by spiral waves appropriate for this study.

Line 100: the claim that thick tissue has formed is at least debatable. I recommend to insert data on the mean \pm -SD or SEM thickness for clarity.

•**Response:** We have added the mean \pm -SD data in Fig 1. Moreover, the sentence has been modified to:

Line 96:

“We created 3D SOTRs by plating hiPSC-CMs in a culture dish with a pillar in the center, around which the CMs aggregated and formed a thick tissue ring within 2 days ($432.72 \pm 56.18 \mu\text{m}$ on day 2, Fig. 1a–c and Supplementary Fig. 1).”

Line 254: the term ring organoid is misleading. An organoid cannot be composed of only cardiomyocytes. The authors are describing a classical aggregation culture. Which is in this case controlled by the central pillar. Similar models have been around for many years.

•**Response:** We have accordingly modified the sentence to:

Line 291:

“This methodology could be an important complement to current widely used methods[15-18], and the rapid formation of the tissue ring without using any exogenous ECM such as collagen or fibrin, might provide new insights to future mass production of matured tissues for drug-screening and

regenerative

applications[6]"

Lines 284/285: must be delete or rephrased; there is no evidence of frequency depended maturation; the TWs are very likely a phenomenon of better cardiomyocyte arrangement, as demonstrated by the FF study.

•**Response:** To make the description more accurate, we have rephrased to sentence to:

Line 329:

“Moreover, the TWs promoted structural and functional maturation of the CMs, thereby offering a supplementary approach to electrical-stimulation-based maturation of electrically active cell types.”

Reviewer #3

In their manuscript “Rapid pacing by circulating traveling waves improves maturation of hiPSC- derived cardiomyocytes in self-organized tissue ring”, Li and colleagues report the creation of 3D self-organized tissue rings (SOTRs) from human induced pluripotent stem cell-derived cardiomyocytes (hiPSC-CMs). Using calcium imaging (as a surrogate for action potential activity), the authors demonstrate different types of spontaneous and sustained traveling waves (TWs) within SOTRs, with frequencies of up to 4Hz. After 2 weeks of SOTR culture, the authors show that as the numbers of TWs increases, anisotropic structural organization increases, cardiac-specific gene expression increases, calcium-handing properties are enhanced, oxygen- consumption rate increases, and contractile force is

enhanced. The authors also create a mathematical model that agrees with the experimental functional characteristics of TWs in the SOTRs.

In contrast to recent studies that use externally applied electrical stimulation to mature hiPSC- CMs (and hiPSC-CM-derived engineered heart tissues (EHTs)) the authors only study the effects of spontaneous electrical activity on hiPSC-CM maturation. The title of their manuscript implies that “(spontaneous) rapid pacing by circulating traveling waves improves maturation”; however, in this manuscript, the authors only show a correlation but not truly causation. Based on their data, the opposite could be true in that maturation improves rapid pacing by circulating traveling waves.

Therefore, there are several questions/comments that the authors are asked to address to strengthen the manuscript over its present form:

Major Comments:

1. The title “Rapid pacing by circulating traveling waves improves maturation of hiPSC- derived cardiomyocytes in self-organized tissue ring” has not been proven in the manuscript in its present form. At present, the authors can state “Rapid pacing by circulating traveling waves is associated with maturation...”

•**Response:** We thank the reviewer for the comments and suggestions. We have added the data of the ring with different TWs at day 2 after plating (Supplementary Fig. 4). At day 2, the formation of rings and their TW number became stable (3D ring formation compared with initial monolayer formation). As the staining results show, all samples, with or without TW, demonstrated randomly distributed and poorly organized CMs and similar maturation. After two weeks of culture, the rings with TW demonstrated an aligned structure and expressed molecular markers at higher levels than those without TWs (Figure 3). In addition, the results of the simulation model in this manuscript indicated that the initial state of CMs within the SOTR led to different TW numbers in the ring (Figure 2). On another hand, it has been previously reported that the rapid pacing could lead to CM maturation [1, 2]. We consider that the TW with a pacing frequency (~4 Hz), although slower than that used in the previous reports (~6 Hz), could still promote higher maturation of CMs compared to those without pacing. Taken together, these finding indicate that the maturation improvement could be the result of TW training, rather than vice versa.

We thus consider to change the title to a more appropriate title:

“Circulating traveling waves rapidly pace and mature hiPSC- derived cardiomyocytes in self-organized tissue ring.”

Supplementary Figure 4. Representative confocal images of SOTRs with zero, one, or two TWs at day 2. CMs were stained with anti- α -actinin (red); anti- β -MHC (red); anti-TnT2 (green); anti-Vimentin (red) and DAPI (blue).

2. By flow cytometry, the authors show up to 90% differentiation efficiency of hPSC-CMs. However, they do not show the subtype composition (nodal, atrial, ventricular) of their differentiations, either by whole

cell patch clamping and/or voltage dye imaging (e.g. FluoVolt or Di-4-ANEPPS). Variation in the presence of nodal (or even atrial) subtypes, which beat inherently faster than ventricular subtypes, could account for the variation in TW number (0, 1, 2, or 3) in the SOTRs. The authors should characterize the subtype composition of their input hPSC-CMs (by patch clamping and/or voltage dye imaging of action potentials).

•**Response:** We have evaluated the relative ventricular to atrial CM composition in the differentiated CMs in our previous report by FACS and patch clamp [4]. The results indicated that the iPS cells (IMR90-1) derived CMs obtained by the differentiation protocol are composed of mostly ventricular cells (60% for MLC2v-positive and 10% for MLC2v/MLC2a-positive) and few atrial cells (MLC2a) and pace maker cells. As for the 253G1 cells (cell line used in this study) derived CMs, the cell composition is similar. (70% for MLC2v-positive cells, 10% for MLC2v/MLC2a-positive cells, 0.2~0.3% for MLC2a-positive cells, 10% for pace maker cells, Response Figure 3). We would like to mention that since the data for 253G1 cells is similar to our previous report [4], we will include the data in Response Figure 3 in another of project that compares the impact of culture methods (Suspension and Attachment) on differentiation efficiency.

[REDACTED]

3. Related to #2 above the authors should characterize the in situ subtype composition of their SOTRs by voltage dye imaging, preferably from days 0-14 of SOTR culture. Patch clamping is not necessary here.

•**Response:** We have used voltage dye (FluoVolt) to characterize the in situ subtype of cardiac cells, and the action potential curve confirmed that the CMs within the SOTR are mostly ventricular cardiomyocytes (Supplemental Fig. 3).

Supplemental Figure 3. Representative fluorescence images of SOTRs with zero, one, or two TWs at day 6. CMs were stained by FluoVolt voltage dye according to the manufacturer's protocol. The dashed line marks the tissue area in the images. The numbered squares mark the area where the membrane potential was recorded.

4. It is interesting that the authors use 20% FBS and 10 ng/mL BMP-4 in the media used to generate SOTRs. What is their rationale for this?

•**Response:** In our differentiation protocol, the hiPSC cluster is used for differentiation, and the cardiomyocyte clusters need to be dissociated into single cells by using the protease solution treatment for 1h. During this process, the cells could suffer damage. In the following SOTR generating step, the 20% fetal bovine serum, rich in varieties of proteins and growth factors, could help the cells to recover from the damage and to establish cell cell adhesion [19]. After day 2, the FBS is removed from the medium to repress the fibroblast proliferation.

As for the BMP-4 content, we would like to apologize for the mistake, this content is from our previous old method (for both differentiation and tissue formation). At the moment, during the SOTRs generation, such factor is no longer used and no difference is found during the tissue generation. We have modified the corresponding Methods section as following:

Line 358:

“Before cell seeding, single CMs were filtered using a 40- μ m cell strainer (BD Falcon; Becton Dickenson, Franklin Lakes, NJ, USA) and resuspended at a density of 2×10^6 cells/mL in culture medium containing 40% high glucose DMEM (Sigma-Aldrich), 40% IMDM (Sigma-Aldrich), 20% fetal bovine serum (FBS; Gibco, USA), 1% minimum essential medium non-essential amino acid solution (Sigma-Aldrich), 0.1% penicillin-streptomycin (Gibco), 0.5% L-glutamine (Sigma-Aldrich).”

5. Related to #4 above, it is interesting that the authors do not use exogenous ECM (such as collagen I or fibrin, as other groups have commonly used to create engineered heart tissues/organoids), but are still able to get compaction of the hPSC-CMs into SOTRs. As FBS is known to stimulate fibroblast proliferation and migration, it is possible that varying amounts of fibroblasts in the SOTRs are contributing to the different TWs observed. The authors should immunostain for fibroblasts in their SOTRs for at least 1-2 fibroblast markers (e.g. periostin (POSTN gene), DDR2 (DDR2 gene); vimentin (VIM gene); FSP-1 (S100A4 gene).

•**Response:** Before re-plating the cells, the FACS data showed that more than 90% percent of the cells were TnT2-positive (Supplementary Fig. 1b). Additionally, the FBS, rich in varieties of proteins and growth factors, could help the CMs to form ring tissue in the first 2 days of culture, and it was removed from the culture medium from day 2 to avoid proliferation of fibroblasts. In addition, we have added the results of fibroblast staining (Supplementary Fig. 4); there were few fibroblasts in both TW samples and in zero TW samples. We thus consider that, according to the simulation result, the starting point of the CM state could decide the TW number within the SOTRs.

Supplementary Figure 4. Representative confocal images of SOTRs with zero, one, or two TWs at day 2. CMs were stained with anti-TnT2 (green), anti-Vimentin (red), and DAPI (blue).

6. Related to #3 and #5 above, in the authors' RNAseq data, they should also be able to estimate nodal (TBX18, SHOX2, MSX2, TBX2, HCN1, HCN4), atrial (MYH6, SLN, NPPA), and ventricular (MYH7, MYL2, IRX4) subtypes and fibroblast (POSTN, DDR2, VIM, S100A4) presence in the SOTRs. Please report relative gene expression of associated genes. Also, compare to levels from publicly available RNAseq data for fetal and adult hearts or other hiPSC-CM data sets.

•**Response:** We have included the RNAseq data comparing the two TWs group, zero TW group, adult group, and iPSC-derived CMs from other report [20]. When compared SOTR CMs with adult heart cells and CMs from previous report. The SOTR CMs showed lower nodal- (*TBX18*, *SHOX2*, *MSX2*, *TBX2*, *HCN1*, and *HCN4*), atrial-related (*MYH6* and *NPPA*), fibroblast-

related (POSTN, DDR2, VIM) and higher ventricular-related (*MYL7*) gene expression than the previously reported CMs. Both the SOTR CMs of this study and the CMs of the previous report showed lower expression of *MYH7* and *MYL2* but higher *IRX4* expression than the adult CMs.

Supplementary Figure 10. FPKM values of cardiac markers from RNA-sequencing data. The sample types include adult heart, two TWs, zero TW and other two iPSC-derived CMs from a previous report [20]. The markers include nodal (*TBX18*, *SHOX2*, *MSX2*, *TBX2*, *HCN1*, and *HCN4*), atrial (*MYH6*, *SLN*, and *NPPA*) and ventricular (*MYH7*, *MYL2*, and *IRX4*) subtypes and fibroblast-related types (*POSTN*, *DDR2*, *VIM*, and *S100A4*).

7. In their mathematical model, the authors take into account CM coupling to simulate TW propagation. In their RNAseq data, do they see differences in gap junction expression (Cx-40/GJA5; Cx-43/GJA1; Cx-45/GJC1) in the SOTRs with different TWs?

•**Response:** We have investigated the abovementioned gap junction expression. The gap junction in working cardiomyocytes (Cx-43/GJA1) and nodal cells (Cx-45/GJC1) showed upregulated expression after TW training. However, no similar upregulation was observed on Cx-40/GJA5, the gap junction in the atrium [21, 22].

Supplementary Figure 11. The expression of Cx-43/GJA1, Cx-45/GJC1, and Cx-45/GJC1 in different sample groups.

8. Related to #7, do the authors see increased N-cadherin (CDH2) expression, both in the RNAseq data and by immunostaining (which would

be expected to show localization at hiPSC-CM ends of SOTRs with 1, 2 and 3 TWs compared to 0 TWs)?

•**Response:** We have now added the immunostaining and RNAseq data on N-cadherin expression. The SOTRs with one and two TWs did show higher expression of N-cadherin both in immunostaining and the RNAseq data.

Supplementary Figure 12. N-cadherin expression in SOTRs with zero, one, or two TWs at day 14. (a) CMs were stained with anti-N-cadherin (red) and DAPI (blue). (b) The FPKM value of N-cadherin (*CDH2*) in different sample groups.

9. Is the gene expression data in Figure 3g qPCR or RNAseq data? Please clarify.

•**Response:** We apologize for the confusion, the information in legend (qPCR) has been updated.

10.To assess relative ventricular to atrial CM composition in the SOTRs, the authors should report the MHC-beta to alpha ratio (MYH7:MYH6).

•**Response:** Please refer to the response to the comment 2 and 3.

11.As other markers of SOTR maturation, the authors should report caveolin-3 (CAV3), beta1-adrenergic receptor (ADBR1), and beta2-adrenergic receptor (ADBR2) expression levels. ADBR1 should increase in expression compared to ADBR2 over time if maturation is occurring.

•**Response:** We have added the data of CAV3, ADBR1, and ADBR2 expression. The TW groups showed lower expression on CAV3 than the adult group. The ADRB1 and ADRB2 showed improved and reduced expression respectively, in the 2 TW samples compared to the zero TW group (Supplementary Figure 11).

Supplementary Figure 11. The expression of CAV3, ADRB1 and ADRB2 in different sample groups.

12.The authors show the ratio of SOTRs with TWs varies with the PDMS pillar diameter (1, 3, and 5mm diameter). One explanation is that the increased passive stretch in the SOTRs with increasing pillar diameter is causing the increased anisotropic structural organization of hPSC-CMs within the SOTRs. If the authors remove the SOTRs from the pillars at day 4

(or cut them) and then allow them to freely float for 1-2 days, do the TWs decrease?

•**Response:** We thank the reviewer for the suggestions. We have added a possible explanation in the manuscript:

Line 174: “Experimentally, we found that SOTRs with a 3-mm pillar were optimal based on their highest occurrence of TWs (~90% at ≥ 1 TW) (Supplementary Fig. 9), which was probably caused by the passive stretch in the SOTRs with different pillar diameter and their effect on the anisotropic structural organization.”

Moreover, we tried to remove the SOTRs from the pillars and, in most cases, the TW was disturbed and disappeared after removal. If the ring is cut, the TW would immediately disappear because the close-loop path for TW to sustainably travel had been broken.

13. Related to #12, would culturing the SOTRs in the presence of an excitation-contraction decoupler (such as blebbistatin or BDM) decrease passive stretch and also the number of TWs?

•**Response:** We thank the reviewer for the suggestions. In fact, as mentioned previously, the TW would be interrupted in certain conditions, such as breaking the path, replacing the medium with a low-temperature medium, among other conditions. We thus believe the presence of decoupler would very likely interrupt and stop the TWs.

14. To prove causation of rapid pacing by circulating traveling waves improving maturation, the authors would need to point stimulate (not field stimulate) the SOTRs with low TW numbers (e.g. 0 and 1) to assess whether they have similar characteristics of SOTRs with higher TW numbers (e.g. 2 and 3). Understandably, if the authors are not able to do this due to technical and/or time constraints, then the most they can claim is that maturation is associated with increased TW number.

•**Response:** We thank the reviewer for the suggestions. In our revised manuscript (Supplementary video 6), the 0 TW sample (day 14) has been rapidly stimulated with point stimulating, and the TW could be successfully generated and maintained in the SOTRs. The existence of TW would not be the result of maturation. We have also added the data on day 2, where all the zero TW, one TW and two TWs showed similar maturation levels (Supplementary Fig. 4). In the simulation result, the randomly initial beating state caused the different TW state. Taking these results together, we conclude that the CM maturation is the result of TW training.

5. Alternative to #14, the authors could treat the SOTRs with beta-blockers to decrease TWs and beta-agonists to increase TWs and measure whether maturation is affected.

•**Response:** We have added the data on isoproterenol and propranolol treatment to the SOTRs with TW (Response Figure 4). The SOTR showed higher beating rate with higher concentration of isoproterenol, whereas the propranolol treatment decreased the TW beating rate. Neither of the drugs increased or reduced the TW number. It is very likely that the maturation

could be further improved by the addition of isoproterenol because of the higher beating frequency. Since the present study is focus on using the TW, a physical phenomenon, to train the CMs for higher maturity, we would like to use these data in future work.

[REDACTED]

[REDACTED]

[REDACTED]

[REDACTED]

[REDACTED]

[REDACTED]

[REDACTED]

[REDACTED]

[REDACTED]

[REDACTED]

[REDACTED]

[REDACTED]

[REDACTED]

[REDACTED]

Minor Comments

1. General grammatical and spelling errors should be updated/corrected throughout the manuscript.

Response.

•**Response:** We have carefully checked and corrected the spelling and grammatical errors in the manuscript.

- [1] K. Ronaldson-Bouchard, S.P. Ma, K. Yeager, T. Chen, L. Song, D. Sirabella, K. Morikawa, D. Teles, M. Yazawa, G. Vunjak-Novakovic, Advanced maturation of human cardiac tissue grown from pluripotent stem cells, *Nature* 556(7700) (2018) 239-243.
- [2] S.S. Nunes, J.W. Miklas, J. Liu, R. Aschar-Sobbi, Y. Xiao, B. Zhang, J. Jiang, S. Massé, M. Gagliardi, A. Hsieh, Biowire: a platform for maturation of human pluripotent stem cell-derived cardiomyocytes, *Nat Methods* 10(8) (2013) 781-787.
- [3] C.P. Jackman, A.L. Carlson, N. Bursac, Dynamic culture yields engineered myocardium with near-adult functional output, *Biomaterials* 111 (2016) 66-79.
- [4] I. Minami, K. Yamada, T.G. Otsuji, T. Yamamoto, Y. Shen, S. Otsuka, S. Kadota, N. Morone, M. Barve, Y. Asai, A small molecule that promotes cardiac differentiation of human pluripotent stem cells under defined, cytokine-and xeno-free conditions, *Cell Rep.* 2(5) (2012) 1448-1460.
- [5] N. Thavandiran, C. Hale, P. Blit, M.L. Sandberg, M.E. McElvain, M. Gagliardi, B. Sun, A. Witty, G. Graham, M. McIntosh, Functional arrays of human pluripotent stem cell-derived cardiac microtissues, *bioRxiv* (2019) 566059.
- [6] F. Weinberger, K. Breckwoldt, S. Pecha, A. Kelly, B. Geertz, J. Starbatty, T. Yorgan, K.-H. Cheng, K. Lessmann, T. Stolen, Cardiac repair in guinea pigs with human engineered heart tissue from induced pluripotent stem cells, *Science translational medicine* 8(363) (2016) 363ra148-363ra148.
- [7] J. Riegler, M. Tiburcy, A. Ebert, E. Tzatzalos, U. Raaz, O.J. Abilez, Q. Shen, N.G. Kooreman, E. Neofytou, V.C. Chen, Human engineered heart muscles engraft and survive long term in a rodent myocardial infarction model, *Circulation research* 117(8) (2015) 720-730.
- [8] J.L. Smeets, M.A. Allesie, W.J. Lammers, F.I. Bonke, J. Hollen, The wavelength of the cardiac impulse and reentrant arrhythmias in isolated rabbit atrium. The role of heart rate, autonomic transmitters, temperature, and potassium, *Circulation research* 58(1) (1986) 96-108.
- [9] H. Izumi-Nakaseko, Y. Nakamura, T. Wada, K. Ando, Y. Kanda, Y. Sekino, A. Sugiyama, Characterization of human iPS cell-derived cardiomyocyte sheets as a model to detect drug-induced conduction disturbance, *The Journal of toxicological sciences* 42(2) (2017) 183-192.
- [10] P.L. Rensma, M.A. Allesie, W.J. Lammers, F.I. Bonke, M.J. Schalij, Length of excitation wave and susceptibility to reentrant atrial arrhythmias in normal conscious dogs, *Circulation research* 62(2) (1988) 395-410.
- [11] M. Courtemanche, J.P. Keener, L. Glass, A delay equation representation of pulse circulation on a ring in excitable media, *SIAM Journal on Applied Mathematics* 56(1) (1996) 119-142.
- [12] M. Courtemanche, L. Glass, J.P. Keener, Instabilities of a propagating pulse in a ring of excitable media, *Physical Review Letters* 70(14) (1993) 2182.
- [13] C.R. Abbiss, P.B. Laursen, Describing and understanding pacing strategies during athletic competition, *Sports Med.* 38(3) (2008) 239-252.
- [14] <<https://en.wikipedia.org/wiki/Pacing>>.
- [15] A. Mihic, J. Li, Y. Miyagi, M. Gagliardi, S.-H. Li, J. Zu, R.D. Weisel, G. Keller, R.-K. Li, The effect of cyclic stretch on maturation and 3D tissue formation of human embryonic stem cell-derived cardiomyocytes, *Biomaterials* 35(9) (2014) 2798-2808.
- [16] S.S. Nunes, J.W. Miklas, J. Liu, R. Aschar-Sobbi, Y. Xiao, B. Zhang, J. Jiang, S. Massé, M. Gagliardi, A. Hsieh, Biowire: a platform for maturation of human pluripotent stem cell-derived cardiomyocytes, *Nat. Methods* 10(8) (2013) 781-787.
- [17] A.F. Godier-Furnémont, M. Tiburcy, E. Wagner, M. Dewenter, S. Lämmle, A. El-Armouche, S.E. Lehnart, G. Vunjak-Novakovic, W.-H. Zimmermann, Physiologic force-

- frequency response in engineered heart muscle by electromechanical stimulation, *Biomaterials* 60 (2015) 82-91.
- [18] W.-H. Zimmermann, K. Schneiderbanger, P. Schubert, M. Didie, F. Münzel, J. Heubach, S. Kostin, W. Neuhuber, T. Eschenhagen, Tissue engineering of a differentiated cardiac muscle construct, *Circulation research* 90(2) (2002) 223-230.
- [19] G. Gstraunthaler, Alternatives to the use of fetal bovine serum: serum-free cell culture, *ALTEX-Alternatives to animal experimentation* 20(4) (2003) 275-281.
- [20] M.A. Branco, J.P. Cotovio, C.A. Rodrigues, S.H. Vaz, T.G. Fernandes, L.M. Moreira, J.M. Cabral, M.M. Diogo, Transcriptomic analysis of 3D Cardiac Differentiation of Human Induced Pluripotent Stem Cells Reveals Faster Cardiomyocyte Maturation Compared to 2D Culture, *Scientific reports* 9(1) (2019) 9229.
- [21] J.H. Lee, S.I. Protze, Z. Laksman, P.H. Backx, G.M. Keller, Human pluripotent stem cell-derived atrial and ventricular cardiomyocytes develop from distinct mesoderm populations, *Cell Stem Cell* 21(2) (2017) 179-194. e4.
- [22] L. Cyganek, M. Tiburcy, K. Sekeres, K. Gerstenberg, H. Bohnenberger, C. Lenz, S. Henze, M. Stauske, G. Salinas, W.-H. Zimmermann, Deep phenotyping of human induced pluripotent stem cell-derived atrial and ventricular cardiomyocytes, *JCI insight* 3(12) (2018).

Reviewers' comments:

Reviewer #1 (Remarks to the Author):

I'd like to thank the authoring team for taking the time to address the raised questions about this manuscript in a comprehensive and meticulous manner. I believe the revised manuscript merits publication in Nature Comm. Biology.

Reviewer #2 (Remarks to the Author):

As already stated in my original review, I find the paper of interest. The focus should be on the description of the method itself, which may, despite its rather low level of maturation, be useful for drug screening applications. Some of my original concerns remain.

1) A demonstration of an application in drug screening rather than speculations about the role of the reentry phenomena in maturation would strengthen the study.

2) The described phenomena are reentry mechanisms and a sign of tissue immaturity, improper electrical organization, or result from the circular design geometry. The term traveling waves must be replaced by proper nomenclature, i.e., reentry. Along the same line, the term spiral wave is reserved for something completely different in cardiac electrophysiology and must not be confused with the described macro-reentry phenomenon.

3) The statement that the phenomena could be used in other cell types is not supported by any data and must be deleted. Heart, neuronal, and retinal cells are physiologically completely different. It is very unlikely that reentry would occur in other cell types.

4) Any claims as to advanced maturation by the reentry phenomena should be avoided. Many other tissue engineering models are around with a much higher degree of maturation. The expression data is of limited value to support the maturation claim.

5) Lines 135-136: Increasing beating frequency is not a sign of maturation. The reduction of reentry events and not their presence is a sign of maturation.

6) Comparison of individual transcript abundance levels in RNAseq data sets from independent reports are rarely useful. Culture conditions and cell composition in the study by Branco et al. were very likely completely different, making it pretty difficult to draw a conclusion.

7) Lines 259-260: calcium handling properties and ADRB1 gene expression do not necessarily correlate.

Reviewer #3 (Remarks to the Author):

In their revised manuscript "Circulating traveling waves rapidly pace and mature hiPSC-derived cardiomyocytes in self-organized tissue ring", Li and colleagues report the creation of 3D self-organized tissue rings (SOTRs) from human induced pluripotent stem cell-derived cardiomyocytes (hiPSC-CMs). Using calcium imaging (as a surrogate for action potential activity), the authors demonstrate different types of spontaneous and sustained traveling waves (TWs) within SOTRs, with

frequencies of up to 4Hz. After 2 weeks of SOTR culture, the authors show that as the numbers of TWs increases, anisotropic structural organization increases, cardiac-specific gene expression increases, calcium-handing properties are enhanced, oxygen-consumption rate increases, and contractile force is enhanced. The authors also create a mathematical model that agrees with the experimental functional characteristics of TWs in the SOTRs. The authors have also added several data to the original manuscript.

In contrast to recent studies that use externally applied electrical stimulation to mature hiPSC-CMs (and hiPSC-CM-derived engineered heart tissues (EHTs)) the authors only study the effects of spontaneous electrical activity on hiPSC-CM maturation. Their revised title of their manuscript continues to imply that "(spontaneous) circulating traveling waves rapidly pace and mature hiPSC-derived cardiomyocytes"; however, in this revised manuscript, the authors continue to only show a correlation but not truly causation. Based on their updated data, the opposite could be true in that maturation improves rapid pacing by circulating traveling waves. Also, the presence of different cardiomyocyte subtype ratios between SOTRs could also be responsible for the TWs and any maturation differences.

Therefore, there are several questions/comments that the authors are asked to address to strengthen the manuscript over its present form:

Major Comments:

1. As stated above, overall the authors still only show correlation but not causation between TWs and hiPSC-CM/SOTR maturation. Throughout the manuscript, the terms "causes", "improves", "promotes", etc. should be replaced with "is associated with" or something similar with respect to the relationship between TWs and maturation or markers of maturation.
2. The new static vs dynamic cultures show an increase in beating frequency but a drop in the TW occurrence ratio in the dynamic group. As other groups have shown that dynamic culture improves maturation, the drop in TW occurrence suggests that only external electrical stimulation can be used in conjunction with dynamic culture, limiting the utility of the TW phenomenon associated with CM maturation.
3. The additional RNA-sequencing data is informing as to the composition of the SOTRs. However, there seems to be no clear pattern of expression between the 0, 1, and 2 TWs compared to adult heart and iPSC-CM controls. It is surprising that some values for iPSC-CMs are significantly higher than the adult heart (e.g. TBX18, SHOX2, MSX2, TBX2, HCN1, HCN4, MYH6, NPPA, IRX4, POSTN, DDR2, VIM, S100A4). This may be due to normalization effects. The authors are encouraged to use Z-score and/or TPM instead of FPKM.
4. The Fluovolt data is a good addition. However, it is surprising that ventricular cells can beat up to 4Hz as shown with 2 TWs without the presence of driving nodal or even atrial cells. In the videos, there seems to be focal points where beating originates. Are the voltage waveforms at the focal points different than at other locations of the SOTRs? In other words, is there a cluster of nodal cells driving the TW rate?
5. Although the increase in ADRB1 compared to ADRB2 increases in the RNAseq data, indicating SOTR maturation, it is still not proven that TWs cause the maturation, but only shows correlation.
6. The Response Figure 4 shows the SOTRs responding as expected to isoproterenol and propranolol. However, this appears to be an acute experiment. In their other manuscript in preparation, the

authors should perform chronic experiments using these drugs to ascertain the effects of increasing and decreasing beating rates with SOTR maturation.

7. The Response Figure 5 shows the SOTRs responding as expected to E4031. However as with #6 above, this appears to be an acute experiment. In their other manuscript in preparation, the authors should perform chronic experiments using this drug to ascertain the effects of decreasing beating rates with SOTR maturation.

8. As before, to prove causation of rapid pacing by circulating traveling waves improving maturation, the authors would need to point stimulate (not field stimulate) the SOTRs with low TW numbers (e.g. 0 and 1) to assess whether they have similar characteristics of SOTRs with higher TW numbers (e.g. 2 and 3). Understandably, if the authors are not able to do this due to technical and/or time constraints, then the most they can claim is that maturation is associated with increased TW number.

This reviewer looks forward to the authors' responses.

Li et al. ***“Rapid pacing by circulating traveling waves improves maturation of hiPSC-derived cardiomyocytes in self-organized tissue ring”*** Manuscript no. COMMSBIO-19-0780-T

Responses to the comments from the reviewers:

We are very grateful to the reviewers and editors for their comments and suggestions. We have accordingly made a number of modifications to our manuscript. We describe the modifications and provide our responses to reviewers' comments (blue text).

Reviewers' comments:

Reviewer #1 (Remarks to the Author):

I'd like to thank the authoring team for taking the time to address the raised questions about this manuscript in a comprehensive and meticulous manner. I believe the revised manuscript merits publication in Nature Comm. Biology.

•**Response:** We are very grateful to the reviewer for the comments.

Reviewer #2 (Remarks to the Author):

As already stated in my original review, I find the paper of interest. The focus should be on the description of the method itself, which may, despite its rather low level of maturation, be useful for drug screening applications. Some of my original concerns remain.

1) A demonstration of an application in drug screening rather than

speculations about the role of the reentry phenomena in maturation would strengthen the study.

•**Response:** Actually, we have demonstrated in the response letter that the SOTR showed significant response to beta-agonist (isoproterenol), a beta-blocker (propranolol) and a potassium blocker (E4031). Since the present work is more focused on how the reentrant wave is originated and maintained in the closed-loop ring and how the reentrant wave affects the CMs maturation, we consider to organize these drug assessment data in a future work focusing on drug screening with SOTRs. Moreover, we have discussed the possible applications of the reentrant phenomena, including drug assessment, in the discussion part.

2) The described phenomena are reentry mechanisms and a sign of tissue immaturity, improper electrical organization, or result from the circular design geometry. The term traveling waves must be replaced by proper nomenclature, i.e., reentry. Along the same line, the term spiral wave is reserved for something completely different in cardiac electrophysiology and must not be confused with the described macro-reentry phenomenon.

•**Response:** We thank the reviewer for the suggestions, we have modified the nomenclature of traveling wave to reentrant wave (RW) throughout the manuscript and supplementary documents.

3) The statement that the phenomena could be used in other cell types is not

supported by any data and must be deleted. Heart, neuronal, and retinal cells are physiologically completely different. It is very unlikely that reentry would occur in other cell types.

•**Response:** We agree with the reviewer that so far there is no evidence that the reentry could occur in other cell types including neuron and retinal cells. We have thus removed the claim in abstract part and discussion part.

4) Any claims as to advanced maturation by the reentry phenomena should be avoided. Many other tissue engineering models are around with a much higher degree of maturation. The expression data is of limited value to support the maturation claim.

•**Response:** We have modified the title to “Rapid pacing by circulating reentrant waves is associated to maturation of hiPSC-derived cardiomyocytes in self-organized tissue ring”. We have also modified the claim such as “causes”, “improves”, “promotes”.

5) Lines 135-136: Increasing beating frequency is not a sign of maturation. The reduction of reentry events and not their presence is a sign of maturation.

•**Response:** We agree with the referee that the increasing beating frequency is not a sign of maturation. Instead, the higher frequency (~6 Hz) could lead to higher maturation as introduced in previous report [1]. In order to have a clearer claim, we have modified the description to following sentences:

Line 133-135:

“which significantly increased from 3.30 ± 0.39 Hz to 3.89 ± 0.18 Hz at day 6, and from 3.92 ± 0.69 Hz to 5.57 ± 0.06 Hz at day 14. The higher beating frequency has been previously reported to be associated with higher maturation of CMs [1].”

6) Comparison of individual transcript abundance levels in RNAseq data sets from independent reports are rarely useful. Culture conditions and cell composition in the study by Branco et al. were very likely completely different, making it pretty difficult to draw a conclusion.

•**Response:** We thank the referee for the comments. As the referee mentioned, the RNAseq data among different reports could be difficult since many experimental conditions varies between different reports. In the future work (e.g., in the ongoing drug screening study with SOTRs), we would compare the 2D and 3D culture from the same cell source and for the same culture periods, from which the data could be more informative and robust.

7) Lines 259-260: calcium handling properties and ADRB1 gene expression do not necessarily correlate.

•**Response:** We thank the referee for the suggestions. The manuscript has been modified accordingly:

Line254-257:

“SOTRs with two RWs exhibited significantly higher-intensity changes than the groups with zero RWs, indicating a higher maturation level in Ca^{2+} -handling properties. Additionally, we recorded the contractile force of SOTRs using a mechanical tester (Fig. 5e and f and Supplementary Fig. 14),”

Reviewer #3 (Remarks to the Author):

In their revised manuscript “Circulating traveling waves rapidly pace and mature hiPSC-derived cardiomyocytes in self-organized tissue ring”, Li and colleagues report the creation of 3D self-organized tissue rings (SOTRs) from human induced pluripotent stem cell-derived cardiomyocytes (hiPSC-CMs). Using calcium imaging (as a surrogate for action potential activity), the authors demonstrate different types of spontaneous and sustained traveling waves (TWs) within SOTRs, with frequencies of up to 4Hz. After 2 weeks of SOTR culture, the authors show that as the numbers of TWs increases, anisotropic structural organization increases, cardiac-specific gene expression increases, calcium-handling properties are enhanced, oxygen-consumption rate increases, and contractile force is enhanced. The authors also create a mathematical model that agrees with the experimental functional characteristics of TWs in the SOTRs. The authors have also added several data to the original manuscript.

In contrast to recent studies that use externally applied electrical stimulation to

mature hiPSC-CMs (and hiPSC-CM-derived engineered heart tissues (EHTs)) the authors only study the effects of spontaneous electrical activity on hiPSC-CM maturation. Their revised title of their manuscript continues to imply that “(spontaneous) circulating traveling waves rapidly pace and mature hiPSC-derived cardiomyocytes”; however, in this revised manuscript, the authors continue to only show a correlation but not truly causation. Based on their updated data, the opposite could be true in that maturation improves rapid pacing by circulating traveling waves. Also, the presence of different cardiomyocyte subtype ratios between SOTRs could also be responsible for the TWs and any maturation differences.

Therefore, there are several questions/comments that the authors are asked to address to strengthen the manuscript over its present form:

Major Comments:

1. As stated above, overall the authors still only show correlation but not causation between TWs and hiPSC-CM/SOTR maturation. Throughout the manuscript, the terms “causes”, “improves”, “promotes”, etc. should be replaced with “is associated with” or something similar with respect to the relationship between TWs and maturation or markers of maturation.

•**Response:** We thank the referee for the suggestions. We have accordingly modified the manuscript. The claim such as “promote” and “improves” have been changed to “is associated with” throughout the manuscript.

Title:

“Rapid pacing by circulating reentrant waves is associated to maturation of hiPSC-derived cardiomyocytes in self-organized tissue ring”

Line 40:

“SOTRs with RWs show higher maturation including structural organization.”

Line 82-83:

“After 2 weeks of culture, the SOTRs with RWs demonstrated improved structural organization”

Line 234-236:

“Collectively, these data, together with the previous day 14 results, indicate that that the RW pacing are associated to improved maturation.”

Line 240:

We removed the sentence “and that faster beating rates during short-term culture could lead to higher levels of maturation”.

Line 243:

“RWs are associated with improved bioenergetics and Ca²⁺-handling properties”

Line 280-283:

“Additionally, although the CMs trained by RWs remained less mature than those generated by state-of-the-art methods[1-3], the maturation level in RWs group could be further affected by changing the stimulation window/duration and improving the beating frequency.”

Line 322:

“Moreover, the RWs are associated with structural and functional maturation of the CMs.”

Legend for Figure 3:

“RWs are associated with upregulation of cardiac-specific gene expression.”

Legend for Figure 4:

“RWs are associated with CM alignment.”

Legend for Figure 5:

“RWs are associated with enhanced Ca^{2+} -handling properties.”

2. The new static vs dynamic cultures show an increase in beating frequency but a drop in the TW occurrence ratio in the dynamic group. As other groups have shown that dynamic culture improves maturation, the drop in TW occurrence suggests that only external electrical stimulation can be used in conjunction with dynamic culture, limiting the utility of the TW phenomenon associated with CM maturation.

•**Response:** As the reviewer mentioned, we do find that after the dynamic culture the beating frequency increase. This could result from dynamic culture that improve the availability of oxygen and glucose to cardiomyocytes in the TW group [4]. And we considered that the drop in the TW occurrence ratio is due to the disturbance caused by the dynamic medium flow (similarly, the TW could disappear during the medium change if the pipetting activities were too fierce). We are currently still working on improve the culture well design and reduce the disturbance of medium during the dynamic culture.

3. The additional RNA-sequencing data is informing as to the composition of the SOTRs. However, there seems to be no clear pattern of expression between the 0, 1, and 2 TWs compared to adult heart and iPSC-CM controls. It is surprising that some values for iPSC-CMs are significantly higher than the adult heart (e.g. TBX18, SHOX2, MSX2, TBX2, HCN1, HCN4, MYH6, NPPA, IRX4, POSTN, DDR2, VIM, S100A4). This may be due to normalization effects. The authors are encouraged to use Z-score and/or TPM instead of FPKM.

•**Response:** We thank the reviewer for the suggestions and comments. We agree with the reviewer that there seems to be no clear pattern between SOTR and the other groups, which could be due to the different culture conditions and cell composition of these different groups. Since the hiPSC-CMs are less mature than the adult CMs, some gene of hiPSC-CMs could be with higher expression level than that of adult ventricular CMs, such like MYH6 [5, 6]. And judging from the low expression level of HCN1 and HCN4 (nodal) and NPPA (atrial) and the high expression of MYH7 (ventricular) in the adult group, one possible hypothesis could be that the adult heart cells specimen (BioChain, USA) are mostly from ventricle of the human heart (the collecting location is not specified in the product datasheet). In addition, the cardiomyocytes differentiated from iPSCs are composed of a mixed population including mostly ventricular, few atrial, nodal cells [7] and fibroblast as demonstrated in our data. This may explain why those nodal, atrial, and fibroblast related genes are upregulated in iPSC-CM compared with those in

adult sample. Moreover, the similar expression differences have been reported in another work [8], which compared cardiomyocytes derived from 9 human iPSC lines and adult heart cells from 11 humans.

4. The Fluovolt data is a good addition. However, it is surprising that ventricular cells can beat up to 4Hz as shown with 2 TWs without the presence of driving nodal or even atrial cells. In the videos, there seems to be focal points where beating originates. Are the voltage waveforms at the focal points different than at other locations of the SOTRs? In other words, is there a cluster of nodal cells driving the TW rate?

•**Response:** Actually, there is no observed spontaneous beating in the SOTR with TW. In the video 1, we have inserted a slowed-down episode showing that the 2 TWs are traveling like two snakes biting each other's tails. In other words, there may not be focal points where beating originates in the SOTR with TWs.

5. Although the increase in ADRB1 compared to ADRB2 increases in the RNAseq data, indicating SOTR maturation, it is still not proven that TWs cause the maturation, but only shows correlation.

•**Response:** We thank the reviewer for the suggestions. As mentioned previously, we have modified the manuscript.

6. The Response Figure 4 shows the SOTRs responding as expected to

isoproterenol and propranolol. However, this appears to be an acute experiment. In their other manuscript in preparation, the authors should perform chronic experiments using these drugs to ascertain the effects of increasing and decreasing beating rates with SOTR maturation.

•**Response:** We thank the reviewer for the suggestions. We will compare the chronic effect by isoproterenol and propranolol and by TWs on the CM maturation.

7. The Response Figure 5 shows the SOTRs responding as expected to E4031. However as with #6 above, this appears to be an acute experiment. In their other manuscript in preparation, the authors should perform chronic experiments using this drug to ascertain the effects of decreasing beating rates with SOTR maturation.

•**Response:** We thank the reviewer for the suggestions. We agree with the reviewer that in the future study, the chronic effect by E4031 should be compared with that by TWs on the CM maturation.

8. As before, to prove causation of rapid pacing by circulating traveling waves improving maturation, the authors would need to point stimulate (not field stimulate) the SOTRs with low TW numbers (e.g. 0 and 1) to assess whether they have similar characteristics of SOTRs with higher TW numbers (e.g. 2 and 3). Understandably, if the authors are not able to do this due to technical

and/or time constraints, then the most they can claim is that maturation is associated with increased TW number.

•**Response:** We thank the reviewer for the comments. As mentioned previously, we have accordingly modified the manuscript.

This reviewer looks forward to the authors' responses.

** See Nature Research's author and referees' website at www.nature.com/authors for information about policies, services and author benefits

COMMSBIO - This email has been sent through the Springer Nature Tracking System NY-610A-NPG&MTS

Confidentiality Statement:

This e-mail is confidential and subject to copyright. Any unauthorised use or disclosure of its contents is prohibited. If you have received this email in error please notify our Manuscript Tracking System Helpdesk team at <http://platformsupport.nature.com> .

Details of the confidentiality and pre-publicity policy may be found here <http://www.nature.com/authors/policies/confidentiality.html>

Privacy Policy | Update Profile

- [1] K. Ronaldson-Bouchard, S.P. Ma, K. Yeager, T. Chen, L. Song, D. Sirabella, K. Morikawa, D. Teles, M. Yazawa, G. Vunjak-Novakovic, Advanced maturation of human cardiac tissue grown from pluripotent stem cells, *Nature* 556(7700) (2018) 239-243.
- [2] D. Zhang, I.Y. Shadrin, J. Lam, H.-Q. Xian, H.R. Snodgrass, N. Bursac, Tissue-engineered cardiac patch for advanced functional maturation of human ESC-derived cardiomyocytes, *Biomaterials* 34(23) (2013) 5813-5820.
- [3] T.J. Herron, A.M.D. Rocha, K.F. Campbell, D. Ponce-Balbuena, B.C. Willis, G. Guerrero-Serna, Q. Liu, M. Klos, H. Musa, M. Zarzoso, Extracellular matrix-mediated maturation of human pluripotent stem cell-derived cardiac monolayer structure and electrophysiological function, *Circulation: Arrhythmia and Electrophysiology* 9(4) (2016) e003638.
- [4] C.P. Jackman, A.L. Carlson, N. Bursac, Dynamic culture yields engineered myocardium with near-adult functional output, *Biomaterials* 111 (2016) 66-79.
- [5] X. Yang, L. Pabon, C.E. Murry, Engineering adolescence: maturation of human pluripotent stem cell-derived cardiomyocytes, *Circulation research* 114(3) (2014) 511-523.
- [6] T. Kamakura, T. Makiyama, K. Sasaki, Y. Yoshida, Y. Wuriyanghai, J. Chen, T. Hattori, S. Ohno, T. Kita, M. Horie, Ultrastructural maturation of human-induced pluripotent stem cell-derived cardiomyocytes in a long-term culture, *Circulation Journal* 77(5) (2013) 1307-1314.
- [7] I. Minami, K. Yamada, T.G. Otsuji, T. Yamamoto, Y. Shen, S. Otsuka, S. Kadota, N. Morone, M. Barve, Y. Asai, A small molecule that promotes cardiac differentiation of human pluripotent stem cells under defined, cytokine-and xeno-free conditions, *Cell Rep.* 2(5) (2012) 1448-1460.
- [8] B.J. Pavlovic, L.E. Blake, J. Roux, C. Chavarria, Y. Gilad, A comparative assessment of human and chimpanzee iPSC-derived cardiomyocytes with primary heart tissues, *Scientific reports* 8(1) (2018) 15312.

REVIEWERS' COMMENTS:

Reviewer #2 (Remarks to the Author):

I have no further comments.

Reviewer #3 (Remarks to the Author):

In their re-revised manuscript, the authors have adequately addressed my comments and questions. Therefore, their manuscript can now be published in its current revised form in Communications Biology.